# Non-autophagic Golgi-LC3 lipidation facilitates TFE3 stress response against Golgi dysfunction

Jaemin Kang [1], Cathena Meiling Li [1], Namhoon Kim[2], Jongyeon Baek[1] & Yong-Keun Jung [1,2]✉

## Abstract

**Lipidated ATG8/LC3 proteins are recruited to single membrane compartments as well as autophagosomes, supporting their functions. Although recent studies have shown that Golgi-LC3 lipidation follows Golgi damage, its molecular mechanism and function under Golgi stress remain unknown. Here, by combining DLK1 overexpression as a new strategy for induction of Golgi-specific LC3 lipidation, and the application of Golgi-damaging reagents, we unravel the mechanism and role of Golgi-LC3 lipidation. Upon DLK1 overexpression, LC3 is lipidated on the Golgi apparatus in an ATG12-ATG5-ATG16L1 complex-dependent manner; a post-Golgi trafficking blockade is the primary cause of this lipidation. During Golgi stress, ATG16L1 is recruited through its interaction with V-ATPase for Golgi-LC3 lipidation. After post-Golgi trafficking inhibition, TFE3, a key regulator of the Golgi stress response, is translocated to the nucleus. Defects in LC3 lipidation disrupt this translocation, leading to an attenuation of the Golgi stress response. Together, our results reveal the mechanism and unexplored function of Golgi-LC3 lipidation in the Golgi stress response.**

**Keywords** CASM; Golgi Apparatus; Golgi Stress Response; Post-Golgi Trafficking; V-ATPase
**Subject Categories** Autophagy & Cell Death; Organelles

## Introduction

In autophagy, ATG8 (mammalian LC3/GABARAPs, hereafter LC3), one of the main regulators of the autophagy process, is recruited to double-membraned autophagosomes after conjugation to phosphatidylethanolamine (PE) (Ichimura et al, 2000; Kabeya et al, 2000). The LC3 lipidation process is mediated by two ubiquitin-like conjugation systems in which ATG12 and LC3 are conjugated to ATG5 and phospholipid, respectively (Mizushima, 2020). ATG7 and ATG10 mediate ATG12-ATG5 conjugation to generate ATG12-ATG5-ATG16L1 complex. After ATG4-mediated proteolytic cleavage of LC3, the ATG12-ATG5-ATG16L1 complex, together with ATG7 and ATG3, conjugates LC3 to phospholipid.

Besides autophagosomes, LC3 also can be recruited to several single membrane structures (Heckmann et al, 2019; Sanjuan et al, 2007; Sønder et al, 2021), known as conjugation of ATG8/LC3 to single membranes (CASM). CASM has distinct molecular mechanisms separate from autophagic LC3 lipidation (Durgan and Florey, 2022). Most of the upstream autophagy regulators, including ULK1/2, FIP200, ATG9, WIPI2, and ATG14L, are dispensable for CASM. The V-ATPase-ATG16L1 axis is a prominent feature of CASM. V-ATPase recruits ATG16L1 to single membrane structures through its interaction with WD40 domain (Fischer et al, 2020; Fletcher et al, 2018; Xu et al, 2019), which is dispensable for autophagy and absent in yeast homolog Atg16 (Mizushima et al, 2003). In autophagy, V-ATPase inhibitors, such as bafilomycin A1 (Baf.A1) and concanamycin A (ConA), increase LC3-II protein levels via lysosomal neutralization (Klionsky et al, 2021). In CASM, however, V-ATPase inhibitors rather suppress LC3 lipidation via dissociation of V0-V1 domains, followed by disruption of V-ATPase-ATG16L1 interaction (Hooper et al, 2022). Compared to PE-specific conjugation in autophagy, LC3 is conjugated to both PE and phosphatidylserine (PS) in CASM (Durgan et al, 2021).

Recent studies have shown that chemical reagents (niclosamide, AMDE-1; autophagy modulator with dual effect-1), fatty acid oleate, and several Golgi-damaging strategies induce LC3 lipidation on the Golgi apparatus (Cerrato et al, 2021; Gao et al, 2016; Gomes-da-Silva et al, 2019; Liu et al, 2019). Golgi-LC3 lipidation also showed unique features of CASM, disengagement of upstream autophagy regulators, and suppression by V-ATPase inhibitors (Gao et al, 2016; Gomes-da-Silva et al, 2019; Liu et al, 2019). However, compared to LC3 lipidation on phagosomes and endolysosomes, the common molecular mechanism and function of Golgi-LC3 lipidation remain largely unknown.

The Golgi apparatus operates as a central hub for modification and intracellular transport of proteins and lipids (Viotti, 2016). Structural or functional disruption of the Golgi apparatus induces Golgi stress, and adaptive cellular responses are accompanied to restore Golgi homeostasis (Kim et al, 2023; Schwabl and Teis, 2022). Golgi apparatus-related degradation (GARD) and endosome and Golgi-associated degradation (EGAD) degrade Golgi proteins by Golgi-associated or cytosolic proteasomes (Eisenberg-Lerner et al, 2020; Schmidt et al, 2019). In Golgi membrane-associated degradation (GOMED), Golgi and cytoplasmic components are engulfed in Golgi-derived double-membraned vesicles, followed by lysosomal degradation (Yamaguchi et al, 2016). Golgiphagy receptors, YIPF3/4 and CALCOCO1, were recently shown to

[1]School of biological sciences, Seoul National University, Seoul 08826, Korea. [2]Interdisciplinary Program in Neuroscience, Seoul National University, Seoul 08826, Korea.
✉E-mail: ykjung@snu.ac.kr

engage in selective degradation of the Golgi apparatus under nutrient stress (Hickey et al, 2023; Kitta et al, 2024; Nthiga et al, 2021). In addition, the "Golgi stress response", composed of TFE3, HSP47, CREB3, MAPK-ETS, proteoglycan, mucin, and PERK pathways, regulates transcription of Golgi-related genes to restore Golgi structure and functions or induce apoptosis (Kim et al, 2023). However, sensors and regulatory mechanisms of most Golgi stress response pathways and the role of LC3 lipidation in Golgi homeostasis have little characterized.

Here, we demonstrate that ectopic expression of Delta-like non-canonical Notch ligand 1 (DLK1), an EGF-like membrane protein transported through the Golgi apparatus, induces non-autophagic Golgi-LC3 lipidation. Golgi-LC3 lipidation results from post-Golgi trafficking blockade and requires two ubiquitin-like conjugation systems and the V-ATPase-ATG16L1 axis, key factors of CASM. Comparing with niclosamide and AMDE-1, we further reveal a common mechanism in that Golgi-LC3 lipidation facilitates TFE3-mediated Golgi stress response to restore Golgi homeostasis.

## Results

### Overexpression of DLK1, not Isoform 2, accumulates LC3 on the *trans*-Golgi network

During a screening process to identify an autophagy regulator using the cDNA expression library encoding membrane proteins, we found that overexpression of DLK1 induced unusual intracellular LC3B accumulation in HeLa cells. DLK1 is a non-canonical Notch ligand which inhibits Notch signaling via interaction with NOTCH1 (Baladrón et al, 2005). As DLK1 is transported to the plasma membrane or secreted through the Golgi apparatus (Macedo and Kaiser, 2019), we hypothesized that the overexpressed DLK1 might recruit LC3B to the Golgi apparatus. Unlike RFP-LC3B distributed in the control cytoplasm, RFP-LC3B highly colocalized with the *trans*-Golgi network membrane protein TGOLN2 (also known as TGN46) upon DLK1 overexpression (Fig. 1A,C). On the other hand, LC3B did not colocalize with the *cis*-Golgi protein GM130 under the same condition (Fig. 1B,D). We confirmed colocalization between DLK1 and TGOLN2-GFP on the *trans*-Golgi network (Figs. 1H and EV1C). Intriguingly, immunoblot analysis showed that DLK1 overexpression increased the conversion of LC3B-I to LC3B-II (Fig. 1E). However, transmission electron microscopy analysis revealed no accumulation of double-membraned autophagosomes on the *trans*-Golgi network (Fig. EV1A). Mammalian LC3 proteins (LC3A, LC3B, and LC3C) and GABARAP proteins (GABARAP, GABARAPL1, and GABARAPL2) were all recruited to the *trans*-Golgi network upon DLK1 over-expression (Appendix Fig. S1). These data suggest that DLK1 overexpression recruits ATG8 proteins to the *trans*-Golgi network.

To examine how DLK1 recruits LC3 to the *trans*-Golgi network, we decided to characterize the region of DLK1. Serial deletion mutants (Δ) of DLK1 lacking one of the six tandem EGF-like repeats, or a cleavage domain and DLK1 Isoform 2 lacking region spanning residue 229 to 301 were generated (Fig. 1F). Among them, overexpression of EGF4-lacking DLK1 mutant and DLK1 Isoform 2 was unable to induce LC3B accumulation (Fig. 1G) and LC3B-II conversion in HeLa cells (Fig. EV1B). When we examined subcellular localization, DLK1 Isoform 2 localized to the *trans*-

Golgi network, while DLK1 mutant lacking EGF-like repeat 4 was found in the ER (Figs. 1H and EV1C). In addition, we examined the overexpression effects of DLL1 and DLL3, Delta-like canonical Notch ligands, which have tandem EGF-like repeats and single transmembrane domains similar to DLK1 (Falix et al, 2012). Unlike DLK1, overexpression of DLL1 or DLL3 neither accumulated LC3B nor induced LC3B-II conversion (Fig. EV1D,E). These results show that a luminal region near the transmembrane domain (TM) is essential for DLK1-specific LC3 accumulation on the *trans*-Golgi network.

### DLK1 expression induces Golgi-LC3 lipidation via ATG12-ATG5-ATG16L1 complex

LC3 lipidation on the membrane compartments in autophagy and CASM is regulated by ATG multiprotein complexes (Durgan and Florey, 2022). To investigate which ATG protein complex regulates the DLK1-induced LC3 lipidation, we observed it in HeLa cells lacking FIP200 or ATG16L1, *Ulk1/2* knockout (KO) mouse embryo fibroblasts (MEFs) and *Atg5* KO MEFs. Loss of Ulk1/2 and FIP200 did not much inhibit the DLK1-induced LC3B accumulation on the *trans*-Golgi network (Fig. 2A,C; Appendix Fig. S2A). On the other hand, loss of Atg5 or ATG16L1, which are all required for LC3 conjugation to phospholipids, completely inhibited the DLK1-induced LC3B accumulation (Fig. 2A,C; Appendix Fig. S2A) and LC3B-II conversion (Fig. 2B). In addition, compared to WT LC3B, LC3B G120A mutant, which replaces glycine of LC3 with alanine and is resistant to the cleavage by ATG4 and thus for the conjugation to phospholipids, was neither accumulated on the Golgi apparatus nor lipidated by DLK1 overexpression (Fig. 2D; Appendix Fig. S2B). Thus, the DLK1-induced Golgi-LC3 lipidation requires ATG12-ATG5-ATG16L1 complex, but not ULK1 complex, providing a new model system of Golgi-specific LC3 lipidation utilizing two ubiquitin-like conjugation systems.

### Inhibition of post-Golgi trafficking precedes Golgi-LC3 lipidation

Although several Golgi-damaging conditions were reported to induce Golgi-LC3 lipidation (Cerrato et al, 2021; Gao et al, 2016; Gomes-da-Silva et al, 2019; Liu et al, 2019), it is not clear yet whether and how LC3 lipidation is associated with different types of Golgi damage. We thus decided to systemically characterize and compare the effects of Golgi-damaging reagents on Golgi-LC3 lipidation, including the most commonly used Golgi damage inducers, brefeldin A and monensin (Dinter and Berger, 1998), and the previously reported Golgi-LC3 lipidation inducers, niclosamide and AMDE-1. As reported, both niclosamide and AMDE-1 triggered robust accumulation of LC3B on the Golgi apparatus in HeLa cells (Fig. 3A). In comparison with niclosamide and AMDE-1, brefeldin A caused dispersion of TGOLN2-RFP and reduced fluorescence of GFP-LC3B and monensin induced partial recruitment of GFP-LC3B to the Golgi apparatus. Therefore, we hypothesized that there may be a specific form of Golgi dysfunction associated with Golgi-LC3 lipidation.

The previous report, in which oleate-induced Golgi-LC3 lipidation is accompanied by protein transport inhibition at the *trans*-Golgi network (Cerrato et al, 2021), led us to investigate whether this protein transport inhibition is a common prerequisite for Golgi-LC3 lipidation. We analyzed protein transport from the

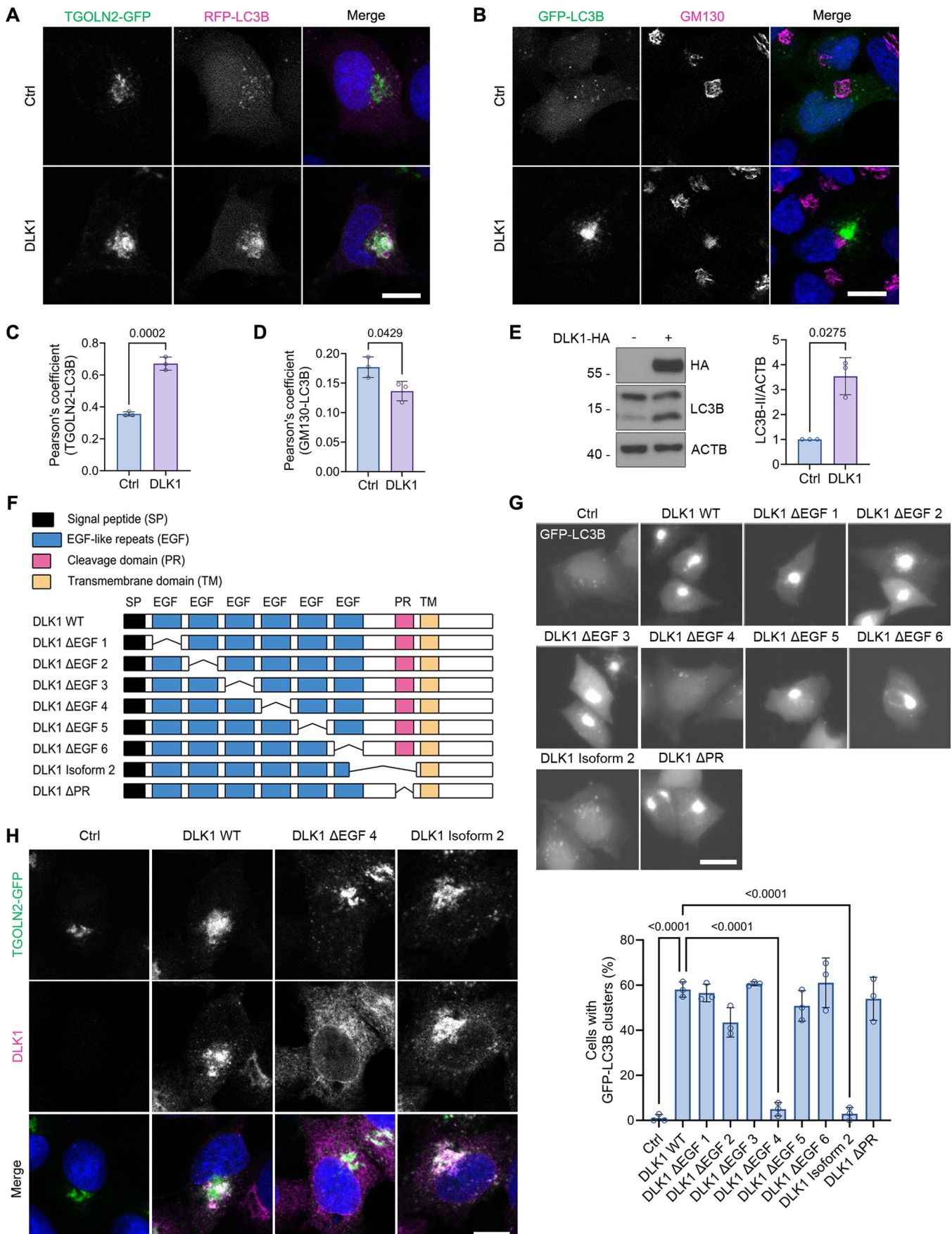

**Figure 1. DLK1 overexpression accumulates LC3 on the *trans*-Golgi network.**

(A, B) Confocal images of HeLa cells expressing DLK1-HA, TGOLN2-GFP, and RFP-LC3B (A) or HeLa cells expressing DLK1-HA and GFP-LC3B and immunostained with anti-GM130 antibody (B). Nuclei were stained by Hoechst dye 33342. Ctrl, control. (C, D) Colocalization of TGOLN2-GFP and RFP-LC3B in (A, C), and of GFP-LC3B and GM130 in (B, D) was quantified with Fiji and represented as Pearson's correlation coefficient. Bars represent mean ± s.d. ($n = 3$, 91–124 cells per experiment in (C) and $n = 3$, 51–92 cells per experiment in (D). Two-tailed Student's $t$-test). (E) Immunoblot analysis of HEK293T cells expressing either pcDNA3-HA (Ctrl) or DLK1-HA (left). Relative signals of LC3B-II to ACTB on the blots are represented as mean ± s.d. ($n = 3$ independent experiments, two-tailed Student's $t$-test) (right). (F) Schematic representation of DLK1 domains, deletion mutants, and isoform. (G) HeLa cells expressing pcDNA3-HA (Ctrl), DLK1-HA WT, or deletion mutants together with GFP-LC3B were observed by fluorescence microscopy (top). The percentages of cells with GFP-LC3B clusters are represented as mean ± s.d. ($n = 3$, 46–97 cells per experiment, one-way ANOVA followed by Tukey's multiple comparisons test, Ctrl vs DLK1 WT; $p = 0.0000000066$, DLK1 WT vs DLK1 ΔEGF4; $p = 0.0000000232$, DLK1 WT vs DLK1 Isoform 2; $p = 0.0000000120$) (bottom). (H) Confocal images of HeLa cells expressing TGOLN2-GFP and either DLK1-HA WT or mutants. Cells were immunostained with anti-DLK1 antibody, and nuclei were stained by Hoechst dye 33342. Scale bars, 10 μm (A, B, H) and 20 μm (G). Source data are available online for this figure.

Golgi apparatus with retention using a selective hooks (RUSH) system (Boncompain et al, 2012). In the RUSH system, a reporter protein fused to streptavidin-binding protein (SBP) is retained in the donor compartment by a hook protein fused to streptavidin (Str), and the addition of biotin releases the reporter protein by interrupting Str-SBP interaction. We utilized Str-KDEL as an endoplasmic reticulum (ER) hook protein and SBP-GFP-GPI as a reporter protein which can be released to the plasma membrane through the Golgi apparatus. As reported, biotin addition released SBP-GFP-GPI from the ER to the plasma membrane through the Golgi apparatus (Fig. 3B,C). In contrast, DLK1 overexpression blocked biotin-induced SBP-GFP-GPI transport from the *trans*-Golgi network, revealed by colocalization between SBP-GFP-GPI and Golgin-97, a *trans*-Golgi network-resident protein (Fig. 3B,D), clearly showing transport inhibition from the *trans*-Golgi network.

We then expanded this analysis to other Golgi-damage inducers and examined their effects on post-Golgi protein transport. As previously shown, brefeldin A and monensin prevented SBP-GFP-GPI secretion from the ER and the Golgi apparatus, respectively (Fig. 3C,E). Consistent with our hypothesis, both niclosamide and AMDE-1 also blocked SBP-GFP-GPI transport from the *trans*-Golgi network (Fig. 3C,E). Some differences in the pattern of the Golgi-LC3 lipidation among monensin, niclosamide, and AMDE-1 might be due to the ionophore function of monensin in other membrane compartments as well as the Golgi apparatus (Grinde, 1983). We further determined whether LC3 lipidation itself prevents post-Golgi trafficking in HeLa cells. Post-Golgi trafficking of SBP-GFP-GPI was also inhibited by niclosamide in HeLa ATG16L1 KO cells as well as WT cells (Fig. 3F), revealing that LC3 lipidation occurs after the trafficking inhibition at the Golgi apparatus. Collectively, these data suggest that post-Golgi trafficking inhibition could be the common cause of Golgi-LC3 lipidation.

## Golgi-LC3 lipidation does not induce autophagic degradation of the Golgi membranes

To investigate whether Golgi apparatus is degraded by autophagy after LC3 lipidation, we generated tandem fluorescent-tagged TGOLN2 (TGOLN2-RFP-GFP), similar to autophagy assay using tandem fluorescent-tagged LC3 in which only GFP fluorescence disappears in acidic compartments (Kimura et al, 2007). If Golgi-derived vesicles are degraded by autophagy, RFP-only dots would be observed since RFP-GFP is conjugated to the cytosolic domain of TGOLN2. We observed the fate of TGOLN2-RFP-GFP under niclosamide incubation and after its removal. Niclosamide treatment rather decreased the number of RFP-only TGOLN2 dots,

which seems to result from the post-Golgi trafficking inhibition (Fig. EV2A). In its wash-out conditions, the number of RFP-only dots did not increase compared to vehicle treatment (Fig. EV2A). Immunoblot analysis revealed that levels of *trans*-Golgi network proteins, TGOLN2 and Golgin-97, were not changed by treatment with and wash-out of niclosamide and AMDE-1 (Fig. EV2B). While the amount of TGOLN2 protein increased with chloroquine treatment during wash-out for 16 h, this increase is likely due to the inhibition of TGOLN2 turnover in basal conditions rather than the effects of niclosamide and AMDE-1 (Fig. EV2C). These results indicate that Golgi-LC3 lipidation is not related to autophagic degradation of the Golgi apparatus.

## The V-ATPase-ATG16L1 axis is essential for Golgi-LC3 lipidation

V-ATPase inhibitors have previously been shown to inhibit Golgi-LC3 lipidation (Gao et al, 2016; Gomes-da-Silva et al, 2019; Liu et al, 2019). We thus examined whether DLK1-induced Golgi-LC3 lipidation could also be reversed by V-ATPase inhibitors. Both V-ATPase inhibitors, Baf.A1 and ConA, abrogated the DLK1-induced LC3B accumulation and LC3B-II conversion in HeLa cells (Figs. 4A and EV3A,C). By contrast, chloroquine (CQ), a lysosomotropic agent which inhibits lysosomal acidification through protonation (Homewood et al, 1972), did not affect it (Fig. EV3B,D). We also found that DLK1 overexpression promoted ATG16L1 recruitment to the *trans*-Golgi network and this recruitment was inhibited by Baf.A1 (Fig. 4B). Next, to understand its mechanism, we tested an interaction between ATG16L1 and ATP6V1A, one of the V1 subunits, in HEK293T cells exposed to niclosamide and AMDE-1, or overexpressing DLK1 (Fig. 4C,E). Immunoprecipitation assay revealed that FLAG-ATG16L1 is bound to endogenous ATP6V1A. Interestingly, both niclosamide and AMDE-1 significantly increased the ATG16L1-ATP6V1A interaction, while Baf.A1 reduced this interaction (Fig. 4C). Similarly, the ATG16L1-ATP6V1A interaction was increased by DLK1 overexpression and decreased by Baf.A1 (Fig. 4E).

ATG16L1 distinguishes LC3 lipidation compartments through its WIPI2b and FIP200-binding domain (FBD) and WD40 domain (Fletcher et al, 2018; Fujita et al, 2008). To determine which domain of ATG16L1 is required for its recruitment to the Golgi apparatus, ATG16L1 deletion mutants (Δ) were generated (Fig. EV3E). Immunoprecipitation assay in HEK293T cells showed that deletion of the WD40 domain, but not FBD, completely impaired the ATG16L1-ATP6V1A interaction upon niclosamide treatment and DLK1 overexpression (Fig. 4D,F). Further, we

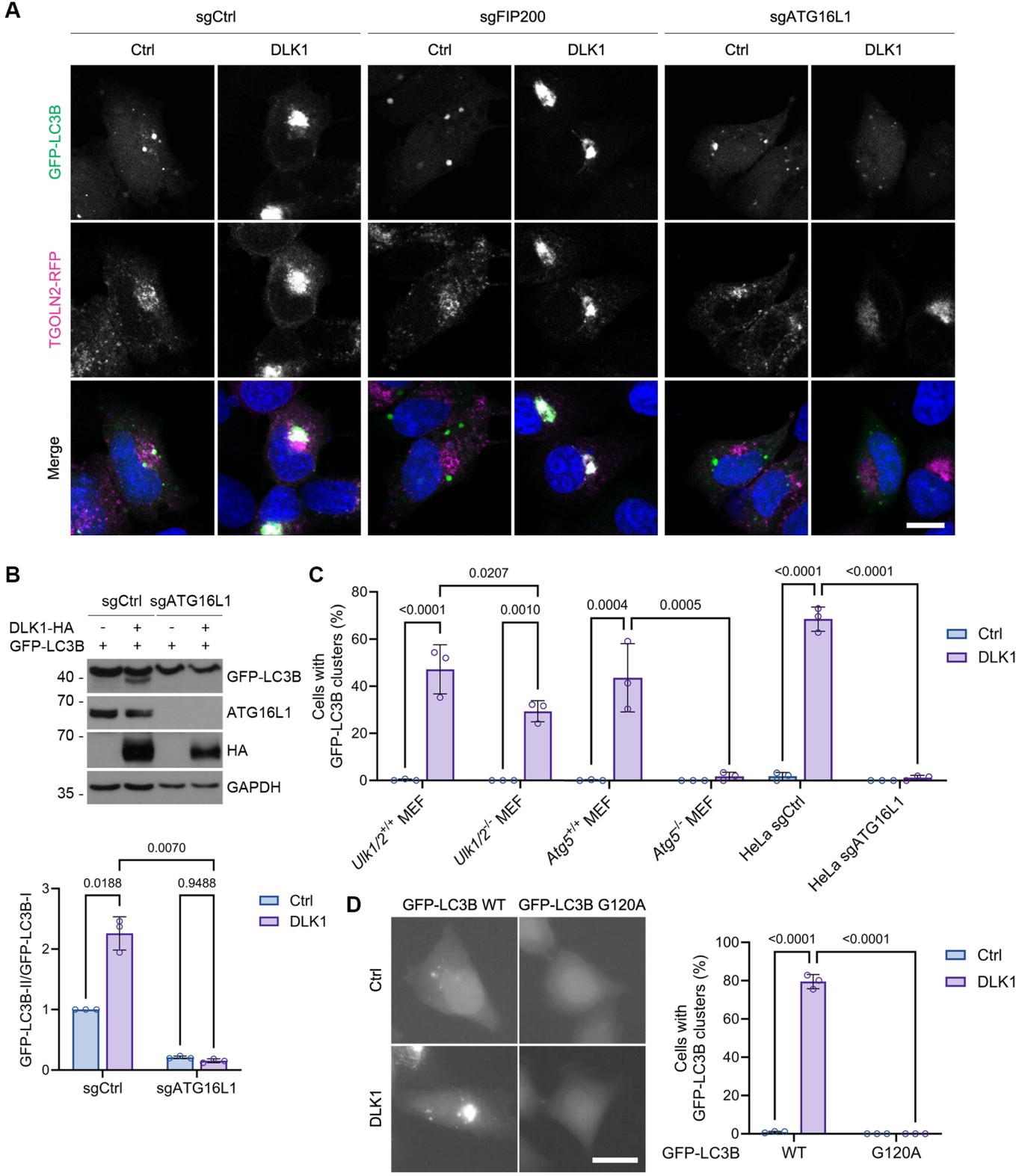

◄

**Figure 2.  LC3 lipidation on the Golgi apparatus by DLK1 expression is ATG12-ATG5-ATG16L1 complex-dependent.**

(A) Confocal images of HeLa sgCtrl, sgFIP200, and sgATG16L1 cells expressing DLK1-HA, GFP-LC3B, and TGOLN2-RFP. Nuclei were stained by Hoechst dye 33342. (B) Immunoblot analysis of HeLa sgCtrl and sgATG16L1 cells expressing GFP-LC3B and either pcDNA3-HA (Ctrl) or DLK1-HA (top). Relative signals of GFP-LC3B-II and GFP-LC3B-I on the blots are represented as mean ± s.d. ($n = 3$ independent experiments, two-way ANOVA followed by Tukey's multiple comparisons test) (bottom). (C) $Ulk1/2^{+/+}$, $Ulk1/2^{-/-}$, $Atg5^{+/+}$, $Atg5^{-/-}$ MEFs, and HeLa sgCtrl, sgATG16L1 cells expressing GFP-LC3B and either pcDNA3-HA (Ctrl) or DLK1-HA were observed by fluorescence microscopy. The percentages of cells with GFP-LC3B clusters are represented as mean ± s.d. ($n = 3$, 84–227 cells per experiment, two-way ANOVA followed by Tukey's multiple comparisons test, $Ulk1/2^{+/+}$/Ctrl vs DLK1; $p = 0.000036$, HeLa sgCtrl/Ctrl vs DLK1; $p = 0.000000011$, HeLa sgCtrl/DLK1 vs sgATG16L1/DLK1; $p = 0.000000011$). (D) Fluorescence microscopy images of HeLa cells expressing DLK1-HA and either GFP-LC3B WT or GFP-LC3B G120A (left). The percentages of cells with GFP-LC3B clusters are represented as mean ± s.d. ($n = 3$, 115–225 cells per experiment, two-way ANOVA followed by Tukey's multiple comparisons test, GFP-LC3B WT/Ctrl vs DLK1; $p = 0.000000000093$, GFP-LC3B WT/DLK1 vs GFP-LC3B G120A/DLK1; $p = 0.000000000079$) (right). Scale bars, 10 µm (A) and 20 µm (D). Source data are available online for this figure.

examined the recruitment of ATG16L1 deletion mutants to the Golgi apparatus. Because ATG16L1 dimerizes mainly via its coiled-coil domain (CCD) (Gammoh, 2020), we expressed ATG16L1 mutants in HeLa ATG16L1 KO cells to exclude the putative dimerization of exogenous RFP-ATG16L1 with endogenous ATG16L1. Except ATG16L1 ΔCCD, we confirmed similar expression levels of ATG16L1 mutants in HeLa ATG16L1 KO cells (Fig. EV3F). Loss of ATG5-binding domain and WIPI2b/FIP200-binding domain did not affect the recruitment of ATG16L1 to the Golgi apparatus by DLK1 (Fig. 4G). In contrast, the loss of the WD40 domain abolished colocalization between ATG16L1 and TGOLN2. This colocalization was reduced by the deletion of the coiled-coil domain, yet the possibility that this was due to the relatively low expression of ATG16L1 ΔCCD cannot be ruled out. These results indicate that Golgi damage recruits ATG16L1 to the Golgi apparatus through WD40 domain-dependent interaction with V-ATPase for LC3 lipidation.

Given dysregulation of luminal ion and pH balance is a common cause of diverse CASM types (Durgan and Florey, 2022), we addressed whether Golgi-LC3 lipidation also is coupled to pH imbalance in the Golgi lumen. We generated a pHluorin-based Golgi pH sensor (GALT-pH2) through conjugation of pHluorin2 to 1-82 amino acids of beta-1,4-galactosyltransferase 1 (B4GALT1), allowing pHluorin2 to localize in the Golgi lumen. The pHluorin2 displays biomodal excitation spectrum peaks at 395 and 475 nm and an emission maximum at 509 nm, and its 395/475 nm excitation ratios decrease by acidification (Mahon, 2011). Golgi apparatus localization of GALT-pH2 was confirmed by its colocalization with TGOLN2-RFP in HeLa cells (Fig. EV4A). Measurement of Golgi pH under post-Golgi trafficking inhibition with flow cytometry analysis of GALT-pH2 fluorescence revealed that DLK1 overexpression increased Golgi pH (GALT-pH2 405/488 nm excitation ratios), while DLK1 Isoform 2, which cannot induce Golgi-LC3 lipidation, did not affect Golgi pH (Fig. EV4B). Unlike DLK1 and Baf.A1, neither niclosamide nor AMDE-1 affected Golgi pH (Fig. EV4C). These results suggest that in the Golgi apparatus, unlike other single-membraned structures, pH imbalance is not likely a common prerequisite for the V-ATPase-ATG16L1 axis.

## Loss of ATG16L1 impairs TFE3 nuclear translocation in post-Golgi trafficking inhibition

We next investigated the function of LC3 lipidation during post-Golgi trafficking inhibition. TFE3, a member of the MiT/TFE transcription factors, is one of the Golgi stress response regulators and is translocated from the cytosol to the nucleus by dephosphorylation under Golgi dysfunction (Taniguchi et al, 2015). We examined whether post-Golgi trafficking inhibition affects TFE3 nuclear translocation. Immunostaining of endogenous TFE3 revealed that both niclosamide and AMDE-1 treatment translocated TFE3 to the nucleus (Fig. 5A). Likewise, DLK1 overexpression also induced TFE3 nuclear translocation (Fig. 5B). As seen in lysosomal damage where lipidated LC3 facilitates TFEB nuclear translocation via interaction with the lysosomal calcium channel TRPML1 (Nakamura et al, 2020), we also examined whether LC3 lipidation on the Golgi apparatus affects TFE3 nuclear translocation. Unlike in WT cells, TFE3 nuclear translocation was impaired in HeLa ATG16L1 KO cells upon niclosamide or AMDE-1 treatment (Fig. 5A,C). The DLK1-induced nuclear translocation of TFE3 was also reduced by ATG16L1 depletion (Fig. 5B,D). With the fractionation assay, we further confirmed that the TFE3 level was increased in WT nuclear fraction by those Golgi damage inducers but was impaired in ATG16L1 KO cells (Fig. 5E–H). With Phos-tag gel electrophoresis, we found that niclosamide or AMDE-1 treatment induced TFE3 dephosphorylation, and these mobility shifts were reduced by ATG16L1 depletion (Appendix Fig. S3A). To clarify whether autophagy, rather than LC3 lipidation itself, regulates TFE3 nuclear translocation, we examined TFE3 translocation in autophagy-deficient HeLa FIP200 KO cells. The results revealed that TFE3 nuclear translocation was not affected in HeLa FIP200 KO cells exposed to AMDE-1 (Appendix Fig. S3B). Thus, LC3 lipidation, but not autophagy itself, triggers TFE3 nuclear translocation after dephosphorylation under post-Golgi trafficking inhibition.

According to the iLIR database (Jacomin et al, 2016), human TFE3 protein has a putative LC3-interacting region (LIR) motif. To determine whether LC3 directly interacts with TFE3 to facilitate nuclear translocation, we performed proximity labeling via conjugation of biotin ligase (TurboID) to LC3B. TurboID-LC3B failed to biotinylate TFE3 during Golgi damage induced by DLK1, niclosamide, or AMDE-1 (Appendix Fig. S4A,B), indicating that LC3 does not bind to TFE3. Considering TFE3 dephosphorylation by calcineurin under ER stress (Martina et al, 2016) and calcium efflux by LC3 lipidation under lysosomal stress (Nakamura et al, 2020), we investigated whether calcium efflux from the Golgi apparatus regulates TFE3 nuclear translocation and is associated with LC3 lipidation. Measurement of calcium levels with Fluo-4 AM and flow cytometry analysis revealed that cytosolic calcium levels slightly increased only by niclosamide, but not DLK1 and AMDE-1, in HeLa cells, and calcium release was not affected by ATG16L1 KO (Appendix Fig. S5A,B). Consistent with these results, incubation with calcium chelator BAPTA-AM did not inhibit TFE3

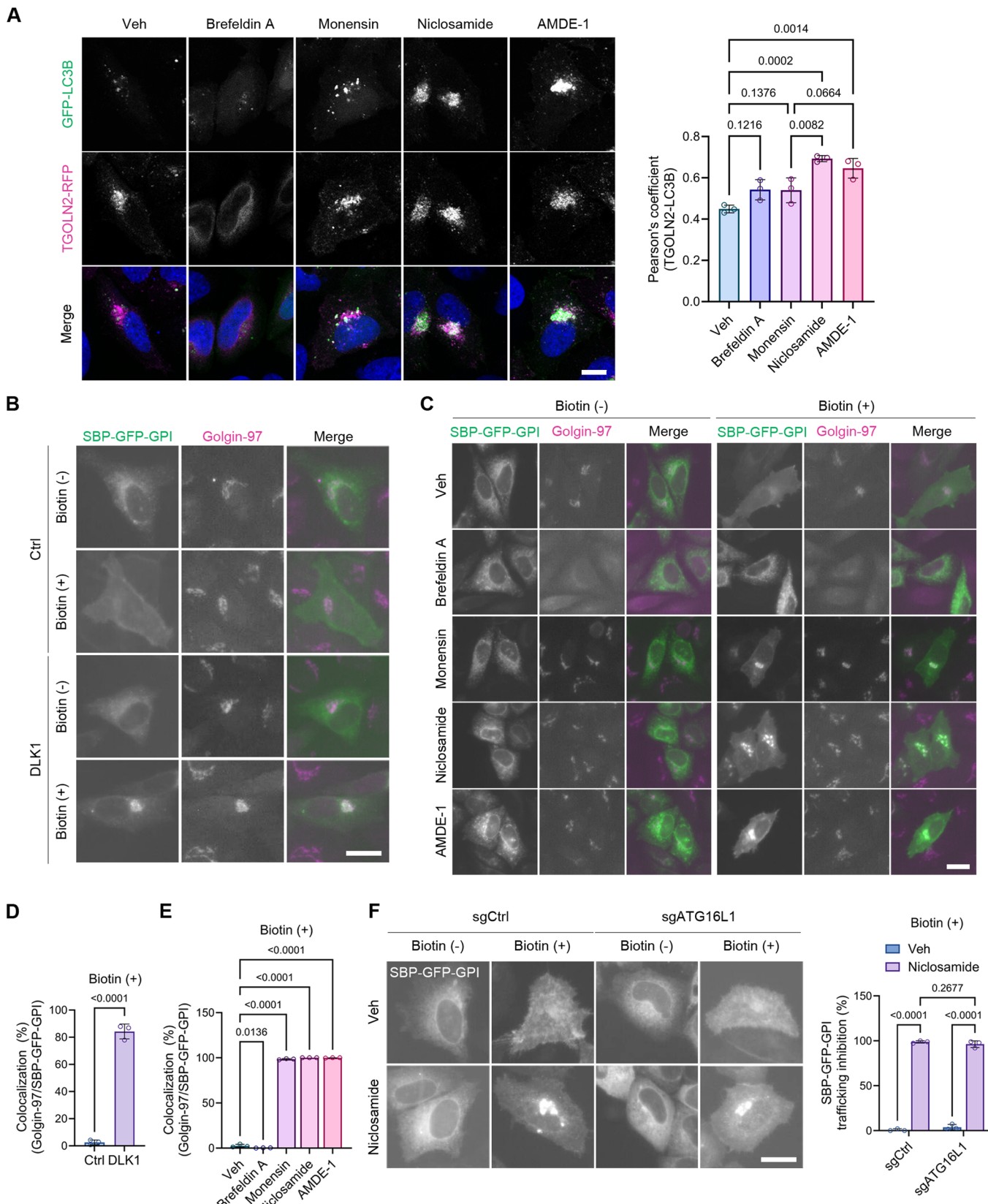

**Figure 3. Post-Golgi trafficking inhibition precedes LC3 lipidation under Golgi stress.**

(A) HeLa cells expressing GFP-LC3B and TGOLN2-RFP were treated with Dimethyl sulfoxide (DMSO, Veh), 5 µg/ml brefeldin A, 5 µM monensin, 10 µM niclosamide, or 10 µM AMDE-1 for 6 h and observed by confocal microscopy. Nuclei were stained by Hoechst dye 33342 (left). Pearson's correlation coefficient of TGOLN2-GFP and RFP-LC3B is represented as mean ± s.d. (n = 3, 53–121 cells per experiment, one-way ANOVA followed by Tukey's multiple comparisons test) (right). (B) HeLa cells expressing DLK1-HA and STR-KDEL_SBP-GFP-GPI were treated with 100 µM biotin, immunostained with anti-Golgin-97 antibody, and observed by fluorescence microscopy. (C) HeLa cells expressing STR-KDEL_SBP-GFP-GPI were treated with DMSO (Veh), 5 µg/ml brefeldin A, 5 µM monensin, 10 µM niclosamide, or 10 µM AMDE-1 for 6 h and treated with 100 µM biotin for additional 2 h. Cells were immunostained with an anti-Golgin-97 antibody and observed by fluorescence microscopy. (D, E) The percentages of cells showing SBP-GFP-GPI trafficking inhibition in (B, D), (C, E), are represented as mean ± s.d. [n = 3, 74–120 cells per experiment, two-tailed Student's t-test, Ctrl vs DLK1; p = 0.000016 (D)], [n = 3, 290–489 cells per experiment, one-way ANOVA followed by Dunnett's multiple comparisons test, Veh vs Monensin; p < 0.000000000000001, Veh vs Niclosamide; p < 0.000000000000001, Veh vs AMDE-1; p < 0.000000000000001 (E)]. (F) HeLa sgCtrl and sgATG16L1 cells expressing STR-KDEL_SBP-GFP-GPI were treated with 10 µM niclosamide for 6 h and treated with 100 µM biotin for additional 2 h. Cells were observed by fluorescence microscopy (left). The percentages of cells showing SBP-GFP-GPI trafficking inhibition are represented as mean ± s.d. (n = 3, 75–194 cells per experiment, two-way ANOVA followed by Tukey's multiple comparisons test, sgCtrl/Veh vs Niclosamide; p = 0.000000000049, sgATG16L1/Veh vs Niclosamide; p = 0.000000000079) (right). Scale bars, 10 µm (A) and 20 µm (B, C, F). Source data are available online for this figure.

nuclear translocation induced by niclosamide and AMDE-1 (Appendix Fig. S5C). Thus, TFE3 regulation following LC3 lipidation under post-Golgi trafficking inhibition is not related to calcium release from the Golgi apparatus.

TFE3 is involved in a wide range of cellular stress responses ranging from nutrient deprivation to various organelle damage, including ER, mitochondria, and lysosomes (Martina et al, 2016; Martina et al, 2014; Nezich et al, 2015; Roczniak-Ferguson et al, 2012). To address whether LC3 lipidation-mediated TFE3 regulation is specific to the Golgi stress, we examined TFE3 nuclear translocation in HeLa ATG16L1 KO cells under several cellular stress conditions. Torin-1, tunicamycin and chloroquine were used to induce mTORC1 inhibition, ER stress, and lysosomal damage, respectively (Fig. EV5A,B). The mTORC1 inhibition by Torin-1 translocated TFE3 to the nucleus, and this translocation was not affected by loss of ATG16L1 in HeLa cells. Loss of ATG16L1 also did not affect the tunicamycin-induced TFE3 nuclear translocation. Unlike Torin-1 and tunicamycin, TFE3 nuclear translocation following lysosomal damage by chloroquine was reduced by ATG16L1 KO in HeLa cells, consistent with the function of LC3 lipidation in the lysosomal damage-induced TFEB activation (Nakamura et al, 2020). Taken together, these data suggest LC3 lipidation facilitates TFE3 activation in a Golgi stress-specific manner.

## LC3 lipidation has a cytoprotective role against the Golgi damage via TFE3 activation

The Golgi stress response pathways restore Golgi homeostasis by upregulation of Golgi-associated target genes (Kim et al, 2023). We thus examined the expression of TFE3 target genes as well as other Golgi stress response pathways under Golgi damage. Consistent to TFE3 nuclear translocation, both niclosamide and AMDE-1 increased mRNA levels of TFE3 target genes (Taniguchi et al, 2015) (SIAT4A, GCP60, Giantin, WIPI49, and STX3A) (Fig. 6A). Niclosamide and AMDE-1 also upregulated pro-apoptotic MCL1 isoform (MCL1-S), the target of MAPK-ETS pathway (Baumann et al, 2018). By contrast, mRNA levels of HSP47 and ARF4, the targets of the HSP47 and CREB3 pathway (Miyata et al, 2013; Reiling et al, 2013), were not affected. DLK1 overexpression also significantly increased mRNA levels of TFE3 target genes and slightly upregulated HSP47, ARF4, and MCL1-S mRNAs (Fig. 6B). Considering the role of TFE3 in the transcription of autophagy and lysosome-related genes, we also examined expression of those genes

under Golgi stress. Compared to the Golgi stress response genes, mRNA levels of autophagy and lysosome-related genes (ATP6V1C1, MCOLN1, CTSA, and ATG16L1) were not significantly increased by niclosamide and AMDE-1 except for ATP6V1C1 (Fig. EV5C). We further addressed whether LC3 lipidation regulates the Golgi stress response genes via TFE3. Gene expression of SIAT4A, GCP60, WIPI49, STX3A, and MCL1-S, which are upregulated in HeLa cells by niclosamide and AMDE-1 was compared with those in ATG16L1 KO cells. Among TFE3 target genes, niclosamide-induced upregulation of WIPI49 and STX3A, vesicle transport-related genes, was significantly inhibited by ATG16L1 knockout (Fig. 6C). Compared to WIPI49 and STX3A, mRNA level of GCP60, implicated in Golgi structure, was less reduced in ATG16L1 KO cells. In addition, expression of SIAT4A, a glycosylation enzyme, was slightly decreased without statistical significance, and MCL1-S in the MAPK-ETS pathway was little affected in HeLa ATG16L1 KO cells.

As TFE3-mediated Golgi stress response has a cytoprotective role against the Golgi damage, we hypothesized that LC3 lipidation has an advantageous role in cell survival under Golgi damages. In HeLa WT cells, niclosamide and AMDE-1 treatment for 24 h showed basal levels of cell toxicity less than 5%, as examined with Calcein-AM and propidium iodide (PI) staining (Fig. 6D). On the other hand, cytotoxicity induced by both chemical reagents significantly increased by the loss of ATG16L1 in HeLa cells. ATG16L1 knockout in HeLa cells also increased cytotoxicity under DLK1 overexpression (Fig. 6E). Together, these data suggest that LC3 lipidation on the Golgi apparatus has a cytoprotective role under Golgi damage through upregulation of the TFE3 pathway target gene expression.

## Discussion

In this study, we have found that Golgi-LC3 lipidation facilitates TFE3 nuclear translocation to alleviate Golgi damage from post-Golgi trafficking inhibition. In addition to the previously identified few chemical inducers of Golgi-LC3 lipidation, we have established a genetic model inducing Golgi-specific LC3 lipidation with DLK1 overexpression. DLK1 primarily performs its physiological functions, including adipogenesis, multiple tissue development, and tumorigenesis, at the plasma membrane (Falix et al, 2012; Pittaway et al, 2021). Among the Notch ligands, we found that overexpressed DLK1 only accumulates largely on the Golgi apparatus, which

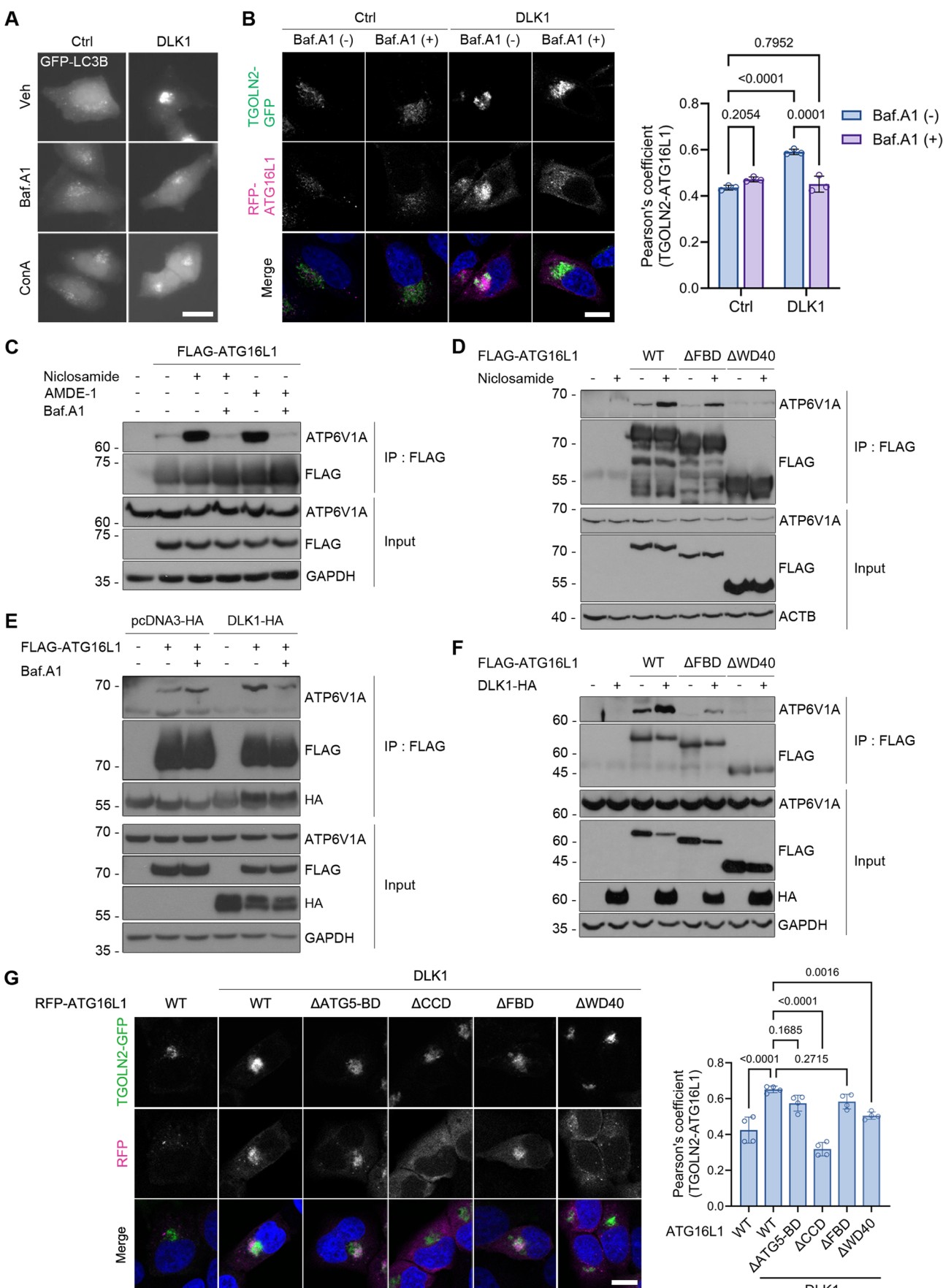

◄ **Figure 4.   The V-ATPase-ATG16L1 axis is required for LC3 lipidation under Golgi stress.**

(A) HeLa cells expressing DLK1-HA and GFP-LC3B were treated with 20 nM bafilomycin A1 (Baf.A1) or 200 nM concanamycin A (ConA) for 6 h and observed by fluorescence microscopy. (B) HeLa cells expressing DLK1-HA, TGOLN2-GFP, and RFP-ATG16L1 were treated with 20 nM bafilomycin A1 (Baf.A1) for 6 h and observed by confocal microscopy. Nuclei were stained by Hoechst dye 33342 (left). Pearson's correlation coefficient of TGOLN2-GFP and RFP-ATG16L1 is represented as mean ± s.d. ($n = 3$, 63–123 cells per experiment, two-way ANOVA followed by Tukey's multiple comparisons test, Ctrl/Baf.A1 (−) vs DLK1/Baf.A1 (−); $p = 0.000057$) (right). (C) HEK293T cells expressing FLAG-ATG16L1 were incubated with 10 μM niclosamide or 10 μM AMDE-1 for 4 h and treated with 20 nM bafilomycin A1 (Baf.A1) for additional 3 h. Cells were subjected to immunoprecipitation (IP) assay with anti-FLAG M2 Affinity Gel. Immunoprecipitates and whole cell lysates (Input) were analyzed by immunoblotting. (D) HEK293T cells expressing FLAG-ATG16L1 WT or FLAG-ATG16L1 mutants were incubated with 10 μM niclosamide for 6 h and subjected to immunoprecipitation (IP) assay with anti-FLAG M2 Affinity Gel. (E) HEK293T cells expressing FLAG-ATG16L1 with or without DLK1-HA were treated with 20 nM bafilomycin A1 (Baf.A1) for 3 h and subjected to immunoprecipitation (IP) assay with anti-FLAG M2 Affinity Gel. (F) HEK293T cells expressing DLK1-HA with FLAG-ATG16L1 WT or FLAG-ATG16L1 mutants were subjected to immunoprecipitation (IP) assay with anti-FLAG M2 Affinity Gel. (G) Confocal images of HeLa sgATG16L1 cells co-expressing DLK1-HA, TGOLN2-GFP, and RFP-ATG16L1 mutants. Nuclei were stained by Hoechst dye 33342 (left). Pearson's correlation coefficient of TGOLN2-GFP and RFP-ATG16L1 mutants is represented as mean ± s.d. ($n = 4$, 21–67 cells per experiment, one-way ANOVA followed by Tukey's multiple comparisons test, Ctrl/ATG16L1 WT vs DLK1/ATG16L1 WT; $p = 0.000008258$, DLK1/ATG16L1 WT vs ATG16L1 ΔCCD; $p = 0.000000028$) (right). Scale bars, 10 μm (B, G) and 20 μm (A). Source data are available online for this figure.

seems to trigger Golgi stress and LC3 lipidation independently of its physiological functions. We also confirmed that overexpression of other membrane or secretory proteins does not induce LC3 lipidation on the Golgi apparatus. Therefore, this genetic model possibly excludes non-specific effects on other organelles than the Golgi apparatus.

Despite increasing reports, it has not been clearly shown what causes Golgi-LC3 lipidation. Here, we believe that dysfunction in post-Golgi trafficking is a common cause of Golgi-LC3 lipidation. Among several Golgi-LC3 lipidation inducers previously reported, only oleate was shown to inhibit protein secretion from the Golgi apparatus (Cerrato et al, 2021). In our assays employing the RUSH system, post-Golgi trafficking was blocked by DLK1 overexpression or niclosamide and AMDE-1 treatment. Especially, the RUSH assay in ATG16L1 KO cells allowed us to conclude that LC3 lipidation is a consequence, rather than a cause, of post-Golgi trafficking inhibition. However, whether post-Golgi trafficking defects under various Golgi stress conditions are primarily and always associated with Golgi-LC3 lipidation remains elusive.

Our study proposes that Golgi-LC3 lipidation under post-Golgi trafficking defects is a Golgi-specific form of CASM based on the following features. First, two ubiquitin-like conjugation systems, but not upstream autophagy regulators, are indispensable for Golgi-LC3 lipidation. Previously, oleate, niclosamide, and AMDE-1 were all reported to induce Golgi-LC3 lipidation independently of autophagy regulators, including ULK1 and class III phosphatidy-linositol 3-kinase (PI3KC3) complex (Gao et al, 2016; Liu et al, 2019; Niso-Santano et al, 2015). Likewise, DLK1 overexpression also induces ULK1/2 and FIP200-independent Golgi-LC3 lipidation. In contrast, molecular machinery involved in LC3-phospholipid conjugation is essential for all types of Golgi-LC3 lipidation. Second, the V-ATPase-ATG16L1 axis, a key molecular feature of CASM which is less studied in the Golgi apparatus, is also essential for these Golgi-LC3 lipidation. In this, the ATP6V1A-ATG16L1 interaction is triggered by Golgi-LC3 lipidation inducers but is suppressed by bafilomycin A1 and WD40 domain deletion. In contrast to endolysosomes, V0-V1 domain assembly at the Golgi apparatus seems not to be triggered by pH imbalance (Fig. EV4). At this moment, we speculate other possible mechanisms by which post-Golgi trafficking inhibition is perceived and conveyed to the V-ATPase-ATG16L1 axis. Considering the role of phosphatidyli-nositol 4-phosphate (PI(4)P) in V-ATPase localization to the Golgi apparatus (Banerjee and Kane, 2017), post-Golgi trafficking

inhibition of PI(4)P might increase V0-V1 domain assembly at the Golgi apparatus. In addition, actin filaments interact with V1 subunits B2 and C1 to maintain V0-V1 domain assembly at the Golgi apparatus (Serra-Peinado et al, 2016). Thus, another plausible model is that post-Golgi trafficking inhibition might affect actin filaments to mediate Golgi-LC3 lipidation via the V-ATPase-ATG16L1 axis.

Autophagy-independent functions of LC3 lipidation in the endolysosomal systems have been characterized in LC3-associated phagocytosis, endocytosis, entosis, extracellular vesicles cargo loading and secretion (Florey et al, 2011; Heckmann et al, 2019; Leidal et al, 2020; Sanjuan et al, 2007). While the role of Golgi-LC3 lipidation under Golgi damage remains still largely elusive, our study reveals that Golgi-LC3 lipidation facilitates TFE3 activation to restore Golgi homeostasis. We propose that LC3 lipidation may function as one of the sensors in the TFE3-mediated Golgi stress response. However, LC3 lipidation is unlikely a sole sensor causing nuclear translocation of TFE3 as there is also LC3 lipidation-independent TFE3 regulation by brefeldin A and monensin (Taniguchi et al, 2015). Among TFE3 target genes, the mRNA levels of vesicle transport-related genes were more significantly affected by LC3 lipidation. This suggests that the transcriptional regulation of Golgi-related genes by TFE3 could be regulated differently depending on the type of Golgi stress. Until now, the phosphatases responsible for TFE3 dephosphorylation under Golgi stress have not been identified. As TFE3 does not directly bind to LC3 (Appendix Fig. S4), Golgi-LC3 lipidation may activate Golgi stress-specific phosphatases to dephosphorylate TFE3 for its nuclear translocation. Given the importance of calcium in TFEB activation under ER and lysosomal stress (Martina et al, 2016; Nakamura et al, 2020), it is possible that several Golgi-resident metal ions, such as $Mg^{2+}$, $Mn^{2+}$, $Zn^{2+}$, and $Cu^{2+}$ (Kellokumpu, 2019), are related to TFE3 activation. Although Golgi-LC3 lipidation and TFE3 stress response are independent of calcium efflux (Appendix Fig. S5), it is necessary to investigate whether other ions regulate Golgi stress-specific phosphatases.

Structural and functional disturbances of the Golgi apparatus resulting from genetic mutations have been implicated in a range of human diseases, with a significant number of these mutations linked to membrane trafficking from the Golgi apparatus (Liu et al, 2021). Mutations in adapter protein complex-1 (AP-1) subunits responsible for clathrin-coated vesicle formation at the trans-Golgi network and in other post-Golgi trafficking pathways (ARFGEF2,

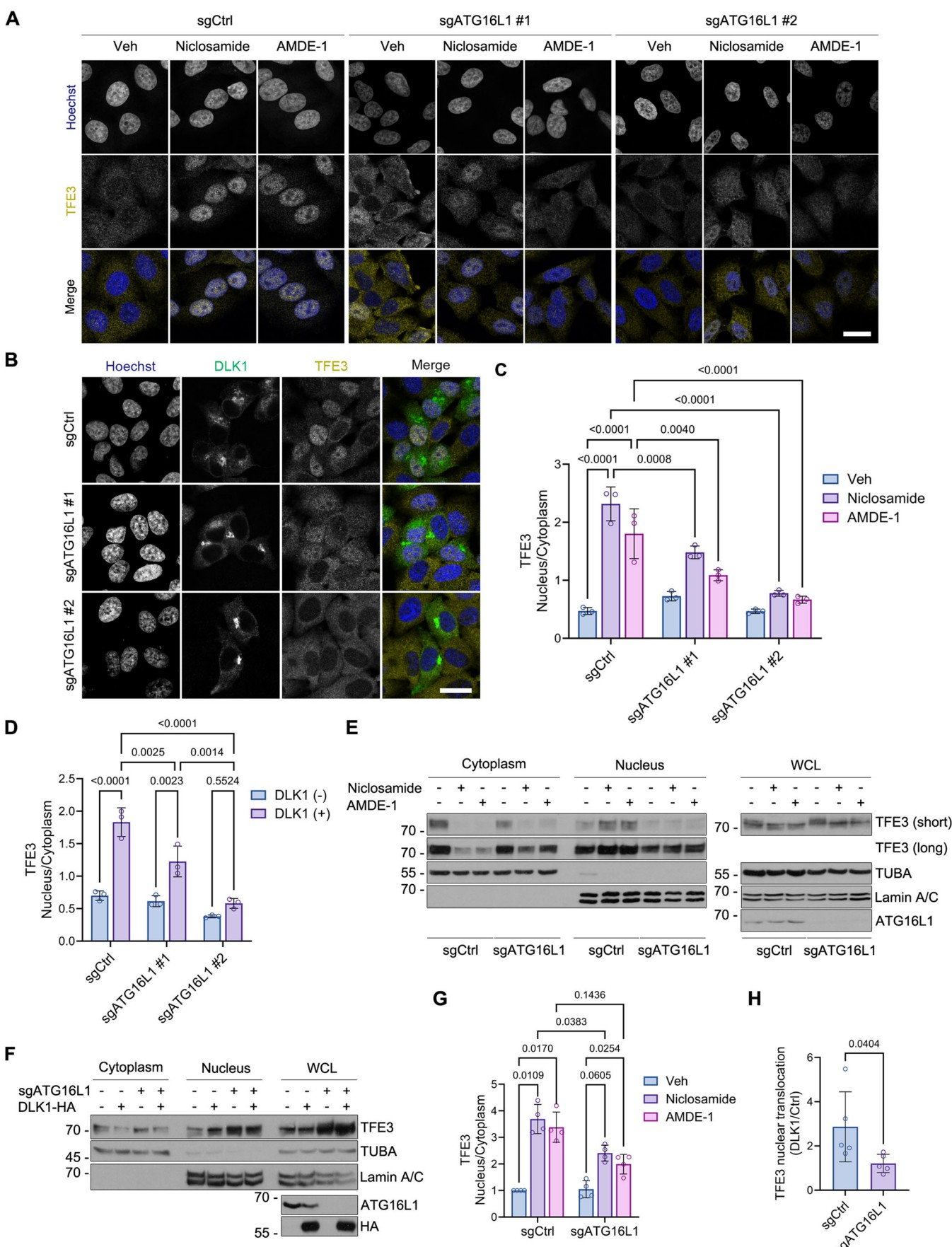

**Figure 5.  LC3 lipidation under Golgi stress facilitates nuclear translocation of TFE3.**

(A) Confocal images of HeLa sgCtrl, sgATG16L1 #1, and sgATG16L1 #2 cells incubated with 10 μM niclosamide or 10 μM AMDE-1 for 6 h and immunostained with anti-TFE3 antibody. Nuclei were stained by Hoechst dye 33342. (B) Confocal images of HeLa sgCtrl, sgATG16L1 #1, and sgATG16L1 #2 cells expressing DLK1-HA and immunostained with anti-DLK1 antibody and anti-TFE3 antibody. Nuclei were stained by Hoechst dye 33342. (C) The nucleus/cytoplasm ratio of TFE3 fluorescence intensity in (A) is represented as mean ± s.d. ($n = 3$, 107–224 cells per experiment, two-way ANOVA followed by Tukey's multiple comparisons test, sgCtrl/Veh vs Niclosamide; $p = 0.000000012$, sgCtrl/Veh vs AMDE-1; $p = 0.000001756$, sgCtrl/Niclosamide vs sgATG16L1 #2/Niclosamide; $p = 0.000000201$, sgCtrl/AMDE-1 vs sgATG16L1 #2/AMDE-1; $p = 0.000016672$). (D) The nucleus/cytoplasm ratio of TFE3 fluorescence intensity in (B) is represented as mean ± s.d. ($n = 3$, 46–247 cells per experiment, two-way ANOVA followed by Tukey's multiple comparisons test, sgCtrl/DLK1 ($-$) vs DLK1 ($+$); $p = 0.0000061$, sgCtrl/DLK1 ($+$) vs sgATG16L1 #2/DLK1 ($+$); $p = 0.0000021$). (E, F) Immunoblot analysis of nuclear and cytoplasmic fractions, and whole cell lysates (WCL) of HeLa sgCtrl and sgATG16L1 cells after incubation with 10 μM niclosamide or 10 μM AMDE-1 for 6 h (E) or expressing pcDNA3-HA (Ctrl) or DLK1-HA (F). (G) Relative signals of nuclear and cytoplasmic TFE3 on the blots in (E) are represented as mean ± s.d. ($n = 4$ independent experiments, two-way ANOVA followed by Tukey's multiple comparisons test). (H) Relative signals of nuclear and cytoplasmic TFE3 between pcDNA-HA and DLK1-transfected cells on the blots in (F) are represented as mean ± s.d. ($n = 5$ independent experiments, two-tailed Student's t-test). Scale bars, 20 μm (A, B). Source data are available online for this figure.

ATP2C1, and ATP7) cause diverse human diseases (García-Cazorla et al, 2022; Liu et al, 2021; Sanger et al, 2019). We hypothesize that Golgi-LC3 lipidation may be induced by genetic mutations associated with post-Golgi trafficking defects. In the future, models of genetic disorders associated with post-Golgi trafficking inhibition need to be established to further elucidate the pathophysiological role of Golgi-LC3 lipidation and TFE3.

In conclusion, we have shown that Golgi stress related to post-Golgi trafficking induces Golgi-LC3 lipidation via the V-ATPase-ATG16L1 axis and have elucidated the function of LC3 lipidation in TFE3 activation. Our findings provide an integrative and in-depth mechanism of Golgi-LC3 lipidation and suggest that LC3 lipidation is a regulator of TFE3 pathway in the Golgi stress response.

# Methods

Reagents including cell lines, plasmids, antibodies, oligonucleotides, chemicals, and software used in this study are listed in the Reagents and Tools Table.

## Cell culture and DNA transfection

HeLa, HEK293T cells, and MEFs were cultured in Dulbecco's modified Eagle's medium (DMEM, Welgene) supplemented with 10% fetal bovine serum (FBS) and 50 μg/ml gentamicin (Gibco) in a 5% $CO_2$ incubator at 37 °C. Cells were tested for mycoplasma contamination by staining with Hoechst dye 33342. Transfection was performed using Lipofectamine 2000 (Thermo Fisher Scientific) or Lipofector-pMAX (AptaBio) according to the manufacturer's instructions.

## Plasmid construction

cDNAs encoding hemagglutinin tag, human DLK1, TGOLN2, TGOLN2-RFP, ATG16L1, LC3B, and GALT were amplified by PCR and inserted into the following vectors, pcDNA3, pcDNA3-HA, pEGFP-N1, mRFP1-N1, mRFP-C1, and pME-pHluorin2. EGFP-LC3B G120A and deletion mutants of DLK1 and ATG16L1 were produced by site-directed mutagenesis. TurboID-LC3B was generated by replacing Sec61b with LC3B from V5-TurboID-Sec61b. All plasmid constructs were verified by DNA sequencing analysis.

## Reagents and tools table

| Reagent/resource | Reference or source | Identifier or catalog number |
| --- | --- | --- |
| **Experimental models** | | |
| HEK293T | ATCC | Cat # CRL-3216 |
| HeLa | ATCC | Cat # CCL-2 |
| HeLa sgCtrl | This study | N/A |
| HeLa sgATG16L1 #1 | This study | N/A |
| HeLa sgATG16L1 #2 | This study | N/A |
| HeLa sgFIP200 | Tsuboyama et al, 2016 | N/A |
| *Ulk1/2*$^{+/+}$ MEF | Cheong et al, 2011 | N/A |
| *Ulk1/2*$^{-/-}$ MEF | Cheong et al, 2011 | N/A |
| *Atg5*$^{+/+}$ MEF | Kuma et al, 2004 | N/A |
| *Atg5*$^{-/-}$ MEF | Kuma et al, 2004 | N/A |
| **Recombinant DNA** | | |
| pcDNA3-HA | This study | N/A |
| pcDNA3-DLK1-HA | This study | N/A |

| Reagent/resource | Reference or source | Identifier or catalog number |
| --- | --- | --- |
| pcDNA3-DLK1-HA ΔEGF1 (Δ24–55) | This study | N/A |
| pcDNA3-DLK1-HA ΔEGF2 (Δ53–86) | This study | N/A |
| pcDNA3-DLK1-HA ΔEGF3 (Δ88–125) | This study | N/A |
| pcDNA3-DLK1-HA ΔEGF4 (Δ135-168) | This study | N/A |
| pcDNA3-DLK1-HA ΔEGF5 (Δ175-205) | This study | N/A |
| pcDNA3-DLK1-HA ΔEGF6 (Δ212-245) | This study | N/A |
| pcDNA3-DLK1-HA Isoform 2 (Δ229-301) | This study | N/A |
| pcDNA3-DLK1-HA ΔPR (Δ278-299) | This study | N/A |
| pCS-MT3-DLL1 | Koo et al, 2005 | N/A |
| pCS-MT3-DLL3 | Koo et al, 2005 | N/A |
| pEGFP-N1-TGOLN2 | This study | N/A |
| pEGFN-N1-TGOLN2-RFP | This study | N/A |
| pEGFP-C1-LC3B | Kabeya et al, 2000 | N/A |
| pEGFP-C1-LC3B (G120A) | This study | N/A |
| mRFP1-N1-TGOLN2 | This study | N/A |
| mRFP1-C1-ATG16L1 | This study | N/A |
| mRFP1-C1-ATG16L1 ΔATG5-BD (Δ13-43) | This study | N/A |
| mRFP1-C1-ATG16L1 ΔCCD (Δ120-206) | This study | N/A |
| mRFP1-C1-ATG16L1 ΔFBD (Δ207-243) | This study | N/A |
| mRFP1-C1-ATG16L1 ΔWD40 (Δ336-607) | This study | N/A |
| mRFP1-C1-LC3B | This study | N/A |
| pcDNA3.1-mRFP-LC3A | Lee et al, 2017 | N/A |
| pcDNA3.1-mRFP-LC3C | Lee et al, 2017 | N/A |
| pcDNA3.1-mRFP-GABARAPL1 | Lee et al, 2017 | N/A |
| pmRFP-C3-GABARAP | Lee et al, 2017 | N/A |
| pmRFP-C3-GABARAPL2 | Lee et al, 2017 | N/A |
| pCMV-3xFLAG-ATG16L1 | Ravikumar et al, 2010 | N/A |
| pCMV-3xFLAG-ATG16L1 ΔFBD (Δ207-243) | This study | N/A |
| pCMV-3xFLAG-ATG16L1 ΔWD40 (Δ336-607) | This study | N/A |
| pcDNA5-V5-TurboID-LC3B | This study | N/A |
| GALT-pHluorin2 | This study | N/A |
| pME-pHluorin2 | Addgene (Mahon, 2011) | Cat # 73794 |
| lentiCRISPR v2 | Addgene (Sanjana et al, 2014) | Cat # 52961 |
| pIRESneo3-Str-KDEL_SBP-EGFP-GPI | Addgene (Boncompain et al, 2012) | Cat # 65294 |
| **Antibodies** | | |
| Mouse monoclonal anti-DLK1 | Santa Cruz Biotechnology | Cat # sc-376755; RRID: AB_3075331 |
| Rabbit polyclonal anti-LC3B | Novus Biologicals | Cat # NB100-2220; RRID: AB_10003146 |
| Rabbit monoclonal anti-ATG16L1 | Cell Signaling Technology | Cat # D6D5; RRID: AB_10950320 |
| Mouse monoclonal anti-Golgin-97 | Thermo Fisher Scientific | Cat # A-21270; RRID: AB_221447 |
| Mouse monoclonal anti-HA | Santa Cruz Biotechnology | Cat # sc-7392; RRID: AB_627809 |
| Mouse monoclonal anti-ACTB | Santa Cruz Biotechnology | Cat # sc-47778; RRID: AB_626632 |
| Mouse monoclonal anti-GAPDH | Santa Cruz Biotechnology | Cat # sc-47724; RRID: AB_627678 |
| Mouse monoclonal anti-TUBA | Santa Cruz Biotechnology | Cat # sc-23948; RRID: AB_628410 |
| Rabbit polyclonal anti-ATP6V1A | Proteintech | Cat # 17115-1-AP; RRID: AB_2290195 |

| Reagent/resource | Reference or source | Identifier or catalog number |
|---|---|---|
| Mouse monoclonal anti-FLAG | Sigma-Aldrich | Cat # F1804; RRID: AB_262044 |
| Rabbit monoclonal anti-TFE3 | Abclonal | Cat # A0548; RRID: AB_2861464 |
| Mouse monoclonal anti-Lamin A/C | Santa Cruz Biotechnology | Cat # sc-376248; RRID: AB_ |
| Mouse monoclonal anti-GM130 | BD Biosciences | Cat # 610822; RRID: AB_398141 |
| Rabbit monoclonal anti-TGOLN2 | Abclonal | Cat # A19618; |
| Mouse monoclonal anti-Myc (9E10) | Santa Cruz Biotechnology | Cat # sc-40; RRID:AB_627268 |
| HRP-conjugated goat anti-mouse IgG, F(ab')$_2$ fragment specific | Jackson ImmunoResearch | Cat # 115-035-006; RRID: AB_2338500 |
| HRP-conjugated goat anti-rabbit IgG, F(ab')$_2$ fragment specific | Jackson ImmunoResearch | Cat # 111-035-006; RRID: AB_2337936 |
| Goat anti-mouse IgG (H + L) secondary antibody, Alexa Fluor 488 | Thermo Fisher Scientific | Cat # A-11001; RRID: AB_2534069 |
| Goat anti-mouse IgG (H + L) secondary antibody, Alexa Fluor 594 | Thermo Fisher Scientific | Cat # A-11005; RRID: AB_2534073 |
| Goat anti-rabbit IgG (H + L) secondary antibody, Alexa Fluor 488 | Thermo Fisher Scientific | Cat # A-11008; RRID: AB_143165 |
| Goat anti-rabbit IgG (H + L) secondary antibody, Alexa Fluor 594 | Thermo Fisher Scientific | Cat # A-11012; RRID: AB_2534079 |
| **Oligonucleotides and other sequence-based reagents** | | |
| sgRNA Ctrl: ACGGAGGCTAAGCGTCGCAA | This study | N/A |
| sgRNA ATG16L1 #1: ACTGAATTACACAAGAAACG | This study | N/A |
| sgRNA ATG16L1 #2: TTGGTGCTTAATCCTCAGTT | This study | N/A |
| qPCR primer SIAT4A | This study | Forward: GGAGGACGACACCTACCGAT Reverse: CCACCGACCTCTTCTCCAG |
| qPCR primer GCP60 | This study | Forward: AGCGTGCATGTCAGTGAGTCC Reverse: GGCACAATCTCATCCAGCAAAG |
| qPCR primer Giantin | This study | Forward: CACTCAGGAGCAGGCACTGTTA Reverse: CAGGACTCGCTTCCATCCAA |
| qPCR primer WIPI49 | This study | Forward: AGTCAGTCACACAAAACCACG Reverse: AGAGCACATAGACCTGTTGGG |
| qPCR primer STX3A | This study | Forward: TCGGCAGACCTTCGGATTC Reverse: TCCTCATCGGTTGTCTTTTTGC |
| qPCR primer HSP47 | This study | Forward: AAGAGCAGCTGAAGATCTGGATG Reverse: GTCGGCCTTGTTCTTGTCAATG |
| qPCR primer ARF4 | This study | Forward: GAGATAGTCACCACCATTCCTACCA Reverse: GGCCTAATTCTATCTTGACCACCA |
| qPCR primer MCL1-S | This study | Forward: GGCCTTCCAAGGATGGGTTT Reverse: ACTCCAGCAACACCTGCAAAA |
| qPCR primer ATP6V1C1 | This study | Forward: GAGTTCTGGCTTATATCTGCTCC Reverse:GTGCCAACCTTTAAGTCAGGAAT |
| qPCR primer MCOLN1 | This study | Forward: TTGCTCTCTGCCAGCGGTACTA Reverse: GCAGTCAGTAACCACCATCGGA |
| qPCR primer CTSA | This study | Forward: CAGGCTTTGGTCTTCTCTCCA Reverse: TCACGCATTCCAGGTCTTTG |
| qPCR primer ATG16L1 | This study | Forward: CAGTTACGTGGCGGCAGGCT Reverse: ACAACGTGCGAGCCAGAGGG |
| qPCR primer GAPDH | This study | Forward: AGAAGGCTGGGGCTCATTTG Reverse: AGGGGCCATCCACAGTCTTC |
| **Chemicals, enzymes, and other reagents** | | |
| Anti-FLAG M2 Affinity Gel | Sigma-Aldrich | Cat # A2220 |
| DMEM | Welgene | Cat # LM 001-05 |
| HBSS | Welgene | Cat # LB 003-04 |

| Reagent/resource | Reference or source | Identifier or catalog number |
|---|---|---|
| Gentamicin | Gibco | Cat #15750078 |
| Dimethyl sulfoxide | Sigma-Aldrich | Cat # D2650 |
| Biotin | Sigma-Aldrich | Cat # B4501 |
| Brefeldin A | Sigma-Aldrich | Cat # B7651 |
| Monensin | Sigma-Aldrich | Cat # M5273 |
| Niclosamide | Sigma-Aldrich | Cat # N3510 |
| AMDE-1 | Key Organics | Cat # 4N-049 |
| Bafilomycin A1 | Sigma-Aldrich | Cat # 196000 |
| BAPTA-AM | Thermo Fischer Scientific | Cat # B1205 |
| Fluo-4, AM | Thermo Fischer Scientific | Cat # F14201 |
| Concanamycin A | Abcam | Cat # ab144227 |
| Chloroquine | Sigma-Aldrich | Cat # C6628 |
| Torin-1 | Tocris Bioscience | Cat # 4247 |
| Tunicamycin | Sigma-Aldrich | Cat # T7765 |
| Lipofectamine 2000 | Thermo Fischer Scientific | Cat # 11668019 |
| Lipofector-pMAX | AptaBio | Cat # AB-LF-M100 |
| Hoechst 33342 | Sigma-Aldrich | Cat # H3570 |
| Phos-tag acrylamide | FujiFilm Wako | Cat # AAL-107 |
| Paraformaldehyde | Sigma-Aldrich | Cat # P6148 |
| Fluoromount aqueous mounting medium | Sigma-Aldrich | Cat # F4680 |
| TRIzol reagent | Molecular Research Center | Cat # TR-118 |
| Moloney murine leukemia virus reverse transcriptase | Enzynomics | Cat # RT001S |
| Oligo (dT)$_{18}$ | GenDEPOT | Cat # 01024-010 |
| Power SYBR Green PCR Master Mix | Applied Biosystems | Cat # 4367659 |
| Propidium iodide | Sigma-Aldrich | Cat # 25535-16-4 |
| Calcein-AM | Thermo Fischer Scientific | Cat # C3099 |
| Protease inhibitor cocktail | Quartett | Cat # QTPPI1012 |
| Sodium orthovanadate | Sigma-Aldrich | Cat # S6508 |
| Sodium fluoride | Sigma-Aldrich | Cat # S1504 |
| Streptavidin-agarose conjugate | Millipore | Cat # 16-126 |
| **Software** | | |
| Fiji | Schindelin et al, 2012 | https://fiji.sc/ |
| GraphPad Prism 10.1.0 | | https://www.graphpad.com/ |
| iLIR | Jacomin et al, 2016 | https://ilir.warwick.ac.uk/ |
| Leica Application Suite X | | https://www.leica-microsystems.com/ |
| **Other** | | |

## Generation of knockout cell lines by CRISPR-Cas9 system

HeLa control cell line and two ATG16L1 knockout cell lines were generated by using independent single guide RNAs (sgRNAs). sgRNA constructs were produced by the insertion of annealed oligonucleotides into lentiCRISPR v2. Each sgRNA was transfected into HeLa cells using Lipofector-pMAX for 24 h, and then cells were incubated with 1 μg/ml puromycin for 48 h. Single clones were isolated by seeding a single cell per well in 96-well plates by serial dilution. Protein expression of ATG16L1 was validated by immunoblotting.

## Cell lysis and immunoblotting

Cells were washed with ice-cold PBS and lysed with RIPA buffer (50 mM Tris-Cl pH 8.0, 150 mM NaCl, 1% NP-40, 0.5% sodium

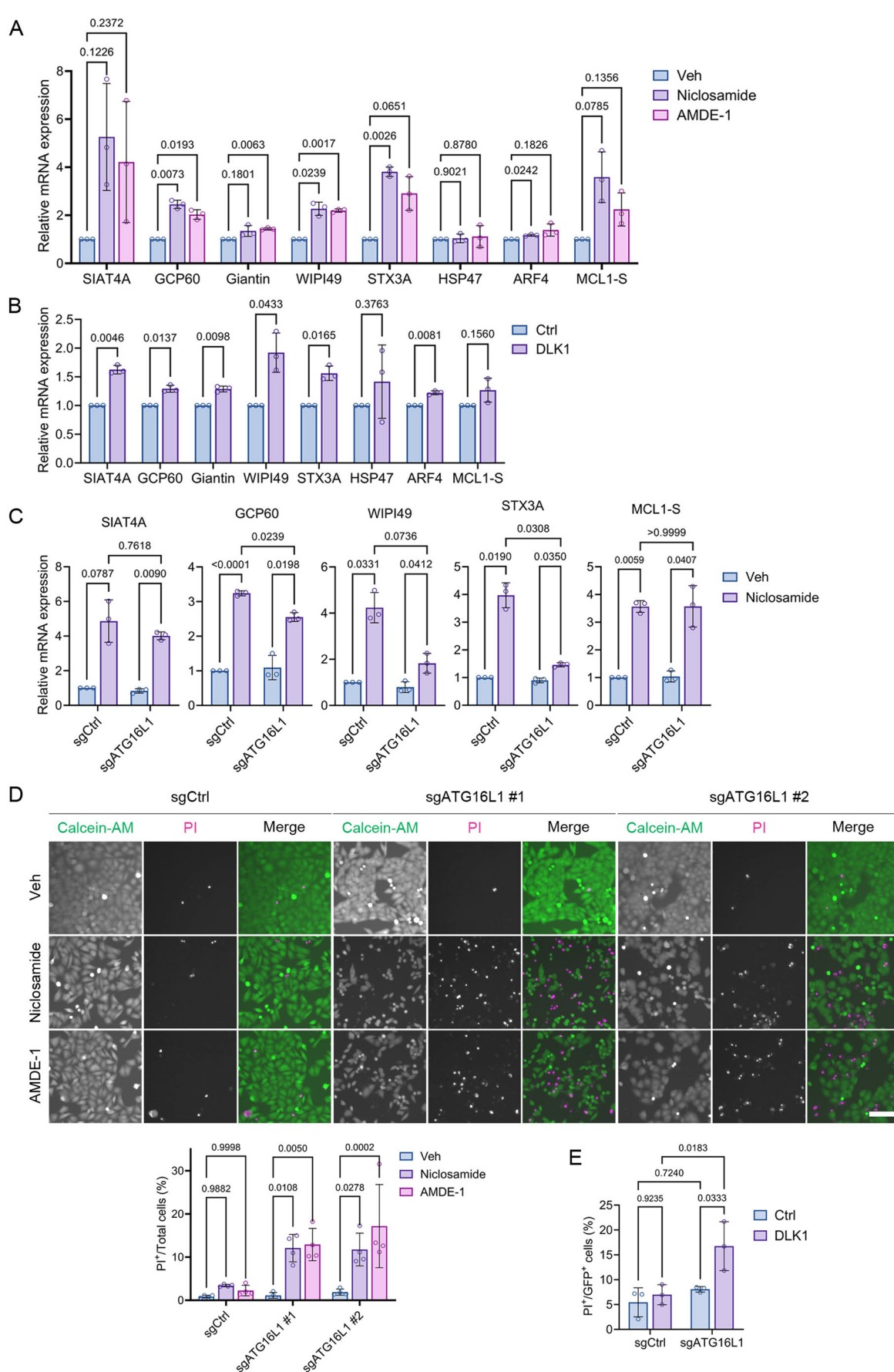

**Figure 6.   LC3 lipidation under Golgi stress ameliorates Golgi apparatus damage via TFE3.**

(A) RNA of HeLa cells exposed to 10 μM niclosamide or 10 μM AMDE-1 for 6 h was analyzed with quantitative real-time PCR. Bars represent mean ± s.d. ($n = 3$ independent experiments, one-way ANOVA followed by Dunnett's multiple comparisons test). (B) RNA of HeLa cells expressing DLK1-HA was analyzed with quantitative real-time PCR. Bars represent mean ± s.d. ($n = 3$ independent experiments, two-tailed Student's *t*-test). (C) RNA of HeLa sgCtrl and sgATG16L1 cells exposed to 10 μM niclosamide for 6 h was analyzed with quantitative real-time PCR. Bars represent mean ± s.d. ($n = 3$ independent experiments, two-way ANOVA followed by Tukey's multiple comparisons test). (D) HeLa sgCtrl, sgATG16L1 #1, and sgATG16L1 #2 cells were treated with 10 μM niclosamide or 10 μM AMDE-1 for 24 h. After incubation with 1 μM Calcein-AM and 5 μM Propidium Iodide (PI), cells were observed by fluorescence microscopy. Scale bar, 100 μm (top). The percentages of PI-positive cells are represented as mean ± s.d. ($n = 4$, 1347–3517 cells per experiment, two-way ANOVA followed by Tukey's multiple comparisons test) (bottom). (E) HeLa sgCtrl and sgATG16L1 cells were co-transfected with DLK1-HA and pEGFP-C1 for 24 h. After incubation with 5 μM propidium iodide (PI), cells were observed by fluorescence microscopy, and the percentages of PI-positive cells are represented as mean ± s.d. ($n = 3$, 493–1255 cells per experiment, two-way ANOVA followed by Tukey's multiple comparisons test). Source data are available online for this figure.

deoxycholate, and 0.1% SDS) supplemented with protease inhibitor cocktail (Quartett) and phosphatase inhibitors (1 mM $Na_3VO_4$ and 1 mM NaF). Supernatants were collected by centrifugation at $12,000 \times g$ for 10 min at 4 °C and protein concentration was measured using the Bradford assay. The supernatants were mixed with 4X Laemmli buffer (250 mM Tris-Cl pH 6.8, 10% SDS, 20% 2-mercaptoethanol, 40% glycerol, 0.08% bromophenol blue) and heated for 10 min at 95 °C. Proteins in cell lysates were separated by SDS-PAGE and transferred to polyvinylidene difluoride (PVDF) membranes. The membranes were blocked with TBST containing 5% bovine serum albumin (BSA) for 1 h at room temperature and incubated with primary antibodies overnight at 4 °C. After being washed three times with TBST, membranes were incubated with secondary antibodies for 1 h at room temperature and further washed four times with TBST. Protein bands were visualized using an enhanced chemiluminescence detection method.

## Immunoprecipitation

Cells were washed with ice-cold PBS and lysed with lysis buffer (50 mM Tris-Cl pH 7.4, 150 mM NaCl, 1 mM EDTA, 1% Triton X-100) supplemented with protease inhibitor cocktail (Quartett) and phosphatase inhibitors (1 mM $Na_3VO_4$ and 1 mM NaF). Supernatants were cleared by centrifugation at $12,000 \times g$ for 10 min at 4 °C and incubated with anti-FLAG M2 Affinity Gel overnight at 4 °C. The gels were washed four times with lysis buffer and eluted with lysis buffer containing 4X Laemmli buffer. After heating for 10 min at 95 °C, the immunoprecipitated proteins were analyzed by immunoblotting.

## TFE3 phosphorylation assay with Phos-tag SDS-PAGE

Proteins in cell lysates were separated by SDS-PAGE supplemented with Phos-tag acrylamide (FujiFilm Wako) and $MnCl_2$ according to the manufacturer's instructions. The gel was washed once in transfer buffer containing 10 mM EDTA for 10 min, followed by washing in transfer buffer without EDTA for 10 min. Proteins were transferred to PVDF membranes. TFE3 proteins were labeled with primary and secondary antibodies and visualized using an enhanced chemiluminescence detection method.

## Immunocytochemistry and microscopy

For immunocytochemistry, cells were seeded on coverslips and transfected with plasmids or incubated with chemical reagents, as indicated in the figures. Cells were fixed with 4% paraformaldehyde

(Sigma-Aldrich) in PBS for 10 min, permeabilized with 0.1% Triton X-100 in PBS for 10 min, and blocked with 4% BSA in PBS-T for 1 h at room temperature. After incubation with primary antibodies diluted in PBS-T with 1% BSA overnight at 4 °C, cells were incubated with secondary antibodies in PBS-T with 1% BSA for 1 h at room temperature. Nuclei were stained with 1 μg/ml Hoechst 33342 (Sigma-Aldrich). Coverslips were mounted using Fluoromount aqueous mounting medium (Sigma-Aldrich). Fluorescently labeled samples were observed by confocal laser scanning microscope (Leica TCS SP8) or fluorescence microscope (Olympus IX-50 equipped with CoolLED pE-300[white] and ProgRes MFcool camera, Jenoptik).

## Image processing and quantification

Images were analyzed with Fiji and processed using LAS X Office software and Adobe Photoshop. Pearson's correlation coefficient was quantified by BIOP JACoP in the Fiji plugin with a region of interest manually drawn around individual cells. To quantify the subcellular localization of TFE3, the nuclear region was determined by Hoechst staining, and the cytoplasmic region was determined by excluding the nuclear region from the entire cellular region. Mean fluorescence intensity of TFE3 in nuclear and cytoplasmic region was measured and the ratio was calculated. For each independent experiment, numerical values obtained from individual cells were averaged, and the mean values were represented as individual symbols on the graphs.

## Retention using selective hooks (RUSH) system

For protein trafficking assay with the RUSH system, HeLa cells were transfected with Str-KDEL_SBP-EGFP-GPI for 24 h and further treated with or without 100 μM biotin for 2 h. Cells were immunostained with anti-Golgin-97 antibody and imaged by fluorescence microscope. The percentages of cells with post-Golgi trafficking inhibition were calculated by dividing the number of cells with EGFP-GPI accumulation at the region of Golgin-97 by the number of EGFP-GPI-expressing cells. In Fig. 3F, the percentages of cells showing EGFP-GPI accumulation at the perinuclear region were calculated to determine post-Golgi trafficking inhibition.

## Nuclear and cytoplasmic fractionation

Cells were lysed in fractionation buffer (20 mM HEPES pH 7.4, 10 mM KCl, 2 mM $MgCl_2$, 1 mM EDTA, 1 mM EGTA, and 1 mM

DTT) supplemented with protease inhibitor cocktail (Quartett) and phosphatase inhibitors (1 mM $Na_3VO_4$ and 1 mM NaF) by suspending through 1 ml syringe with a 26 gauge needle. After 20 min, a portion of cell lysates were collected as whole cell lysate (WCL) fraction and the remaining cell lysates were centrifuged at $720 \times g$ for 5 min at 4 °C. The pellet represented nuclear fraction and was washed twice with fractionation buffer, lysed in RIPA buffer containing 4X Laemmli buffer. The supernatant represented the cytoplasmic fraction and was cleared by centrifugation at $12,000 \times g$ for 10 min at 4 °C, mixed with 4X Laemmli buffer. Cytoplasmic, nuclear fraction, and WCL were analyzed by immunoblotting.

### Flow cytometry for Golgi pH measurement

For Golgi pH measurement, HeLa cells were transfected with GALT-pH2 for 24 h. Cells were trypsinized with trypsin-EDTA and resuspended with Hank's balanced salt solution (HBSS) supplemented with 1.5 mM $CaCl_2$, 10 mM HEPES, 5% fetal bovine serum, and 5.55 mM glucose. Flow cytometry was performed using Flow Activated Cell Sorter canto II. Non-transfected cells were also sorted and the 405/488 excitation ratios were analyzed in cells exhibiting fluorescence signals.

### Flow cytometry for cytosolic calcium measurement

For calcium measurement, cells were transfected with DLK1-HA for 24 h or treated with 10 μM niclosamide or 10 μM AMDE-1 for 6 h. Cells were loaded with 4 μM Fluo-4 AM and 2 mM probenecid (Sigma) for 30 min. Cells were resuspended with Hank's balanced salt solution (HBSS) supplemented with 1.5 mM $CaCl_2$, 10 mM HEPES, 5% fetal bovine serum, 2 mM probenecid, and 5.55 mM glucose. Flow cytometry was performed using Flow Activated Cell Sorter canto II, and the fluorescence intensity at 488 nm was measured.

### RNA extraction and reverse transcription quantitative real-time PCR

Total RNA was extracted using TRIzol reagent (Molecular Research Center) according to the manufacturer's instructions. Complementary DNA was generated using Moloney murine leukemia virus reverse transcriptase (M-MLV RT, Enzynomics) and Oligo $(dT)_{18}$ primer (GenDEPOT). Quantitative real-time PCR was performed using Power SYBR Green PCR Master Mix (Applied Biosystems) on QuantStudio 3 real-time PCR system (Applied Biosystems). Target gene expression was normalized to GAPDH.

### Cell viability assay

Dead cells were stained with 5 μM propidium iodide (PI, Sigma-Aldrich), and viable cells were stained with 1 μM calcein-AM (Thermo Fischer Scientific). In Fig. 6E, cells were co-transfected with pEGFP-C1. Cells were observed by fluorescence microscope. Cell viability was calculated by dividing the number of PI-positive cells by the number of total cells [calcein-AM-positive cells (Fig. 6D) or GFP-positive cells (Fig. 6E) plus PI-positive].

### Proximity labeling experiments

For LC3B-proximal protein labeling under DLK1 overexpression, HeLa cells were transfected with pcDNA3-HA or DLK1-HA together with TurboID-LC3B for 24 h and further treated with 100 μM biotin for 6 h. For LC3B-proximal protein labeling under niclosamide and AMDE-1 incubation, HeLa cells were transfected with TurboID-LC3B for 24 h and treated with 10 μM niclosamide or 10 μM AMDE-1 together with 100 μM biotin for 6 h. Cells were washed three times with ice-cold PBS to stop the labeling reaction and lysed with RIPA buffer supplemented with a protease inhibitor cocktail (Quartett). Supernatants were cleared by centrifugation at $12,000 \times g$ for 10 min at 4 °C and incubated with streptavidin-agarose conjugate (Millipore) overnight at 4 °C. The beads were washed four times with RIPA buffer and eluted with RIPA buffer containing 4X Laemmli buffer. After heating for 10 min at 95 °C, the purified biotinylated proteins were analyzed by immunoblotting.

### Transmission electron microscopy

Cells were fixed with Karnovsky's fixative buffer and washed three times with 0.05 M sodium cacodylate buffer. Postfixation was performed with 1% osmium tetroxide in 0.1 M sodium cacodylate buffer. Cells were washed three times with distilled water, stained with 0.5% uranyl acetate, and further washed three times with distilled water. After serial dehydration with increasing concentration of ethanol, cells were transiently embedded in Spurr's resin. Ultrathin sections were obtained and observed using a JEM1010 transmission electron microscope (JEOL).

### Statistics and reproducibility

All experiments were independently repeated at least three times, and quantitative data were represented as the mean ± standard deviation (s.d.). Statistical analyses were performed using Graph-Pad Prism 10, and calculated $P$ values are described in figures. No statistical methods were used to predetermine the sample sizes. The number of biologically independent experiments and type of statistical test are indicated in the figure legends. Comparison between the two groups was determined by a two-tailed Student's $t$-test. Data from three or more groups with one independent variable were compared by one-way ANOVA followed by Dunnett's or Tukey's multiple comparisons test. Data from two independent variables were compared by two-way ANOVA followed by Tukey's multiple comparisons test. Comparison of relative protein levels from immunoblot data and relative mRNA expression from real-time PCR data was determined by repeated measures of one-way or two-way ANOVA using Geisser-Greenhouse correction, followed by Dunnett's or Tukey's multiple comparisons test.

## Data availability

The source data of this paper are collected in the following database record: biostudies:S-SCDT-10_1038-S44318-024-00233-y.

# Peer review information

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

## Acknowledgements

We would like to thank Craig B. Thompson (Memorial Sloan-Kettering Cancer Center, USA) for *Ulk1/2*$^{+/+}$ and *Ulk1/2*$^{-/-}$ MEFs and Noboru Mizushima (The University of Tokyo, Japan) for *Atg5*$^{+/+}$, *Atg5*$^{-/-}$ MEFs, and HeLa FIP200 knockout cells. We also thank Tamotsu Yoshimori (Osaka University, Japan) for EGFP-LC3B, Jin-A Lee (Hannam University, Korea) for mRFP-tagged LC3/GABARAP constructs, Young-Yun Kong (Seoul National University, Korea) for DLL1-Myc and DLL3-Myc, David C. Rubinsztein (University of Cambridge, UK) for FLAG-ATG16L1, Frank Perez (Institut Curie, France) for Str-KDEL_SBP-EGFP-GPI, Hyun-Woo Rhee (Seoul National University, Korea) for V5-TurboID-Sec61b, David W. Raible (University of Washington, USA) for pME-pHluorin2 and Feng Zhang (Massachusetts Institute of Technology, USA) for lentiCRISPR v2. This work was supported by a grant from the Korea Dementia Research Project through the Korea Dementia Research Center (KDRC), funded by the Ministry of Health & Welfare and Ministry of Science and ICT, Korea [RS-2023-KH134817], and a CRI grant (NRF-2022R1A2B5B03001249) funded by the Ministry of Education, Science and Technology, Korea.

## Author contributions

**Jaemin Kang**: Conceptualization; Data curation; Formal analysis; Supervision; Validation; Investigation; Visualization; Writing—original draft; Writing—review and editing. **Cathena Meiling Li**: Formal analysis; Validation; Investigation. **Namhoon Kim**: Formal analysis; Validation; Investigation. **Jongyeon Baek**: Formal analysis; Investigation. **Yong-Keun Jung**: Conceptualization; Supervision; Funding acquisition; Writing—original draft; Project administration; Writing—review and editing.

Source data underlying figure panels in this paper may have individual authorship assigned. Where available, figure panel/source data authorship is listed in the following database record: biostudies:S-SCDT-10_1038-S44318-024-00233-y.

## Disclosure and competing interests statement

The authors declare no competing interests.

# Expanded View Figures

**Figure EV1. Overexpression of DLK1 Isoform 2, DLL1 and DLL3 does not induce LC3 accumulation on the *trans*-Golgi network.**

(A) Transmission electron microscopy images of HeLa cells expressing pcDNA3-HA (Ctrl) or DLK1-HA. G, Golgi apparatus. Arrowheads indicate single-membraned vesicles associated with a *trans*-Golgi network. (B) Immunoblot analysis of HeLa cells expressing GFP-LC3B and either DLK1-HA WT or deletion mutants (left). Relative signals of GFP-LC3B-II and GFP-LC3B-I on the blots are represented as mean ± s.d. ($n = 5$ independent experiments, one-way ANOVA followed by Dunnett's multiple comparisons test) (right). (C) HeLa cells expressing TGOLN2-GFP and either DLK1-HA WT or mutants were immunostained with anti-DLK1 antibody and observed by confocal microscopy. Pearson's correlation coefficient of TGOLN2-GFP and DLK1 is represented as mean ± s.d. ($n = 3$, 91–125 cells per experiment, one-way ANOVA followed by Dunnett's multiple comparisons test, DLK1 WT vs DLK1 ΔEGF4; $p = 0.000014$). (D) HeLa cells expressing GFP-LC3B with either DLK1-HA, DLL1-Myc, or DLL3-Myc were observed by fluorescence microscopy (left). The percentages of cells with GFP-LC3B clusters are represented as mean ± s.d. ($n = 3$, 71–126 cells per experiment, one-way ANOVA followed by Tukey's multiple comparisons test, Ctrl vs DLK1; $p = 0.000000038$, DLK1 vs DLL1; $p = 0.000000052$, DLK1 vs DLL3; $p = 0.000000049$) (right). (E) Immunoblot analysis of HeLa cells expressing GFP-LC3B with either DLK1-HA, DLL1-Myc, or DLL3-Myc (left). Relative signals of GFP-LC3B-II to GFP-LC3B-I on the blots are represented as mean ± s.d. ($n = 3$ independent experiments, one-way ANOVA followed by Tukey's multiple comparisons test) (right).

▶

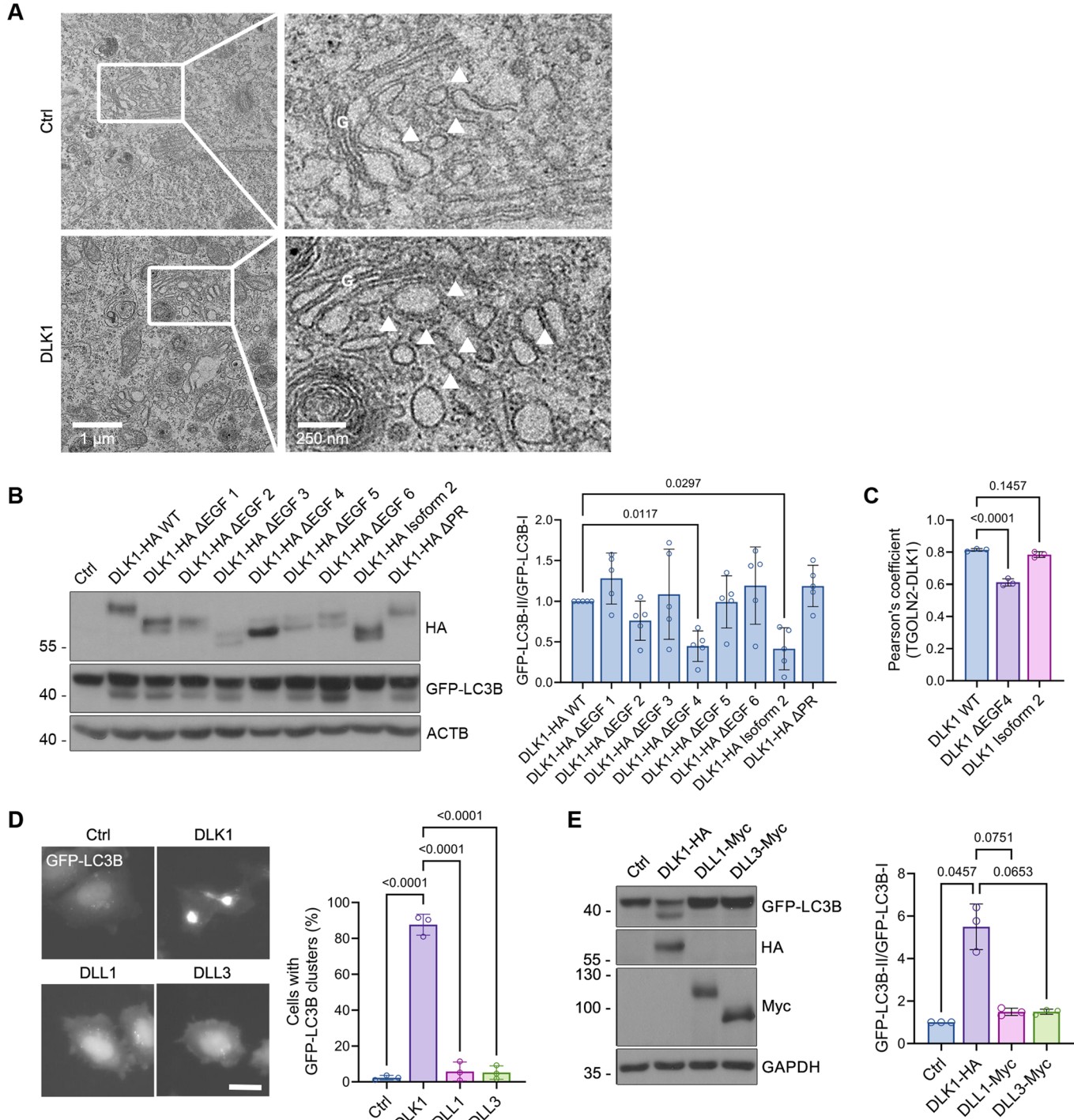

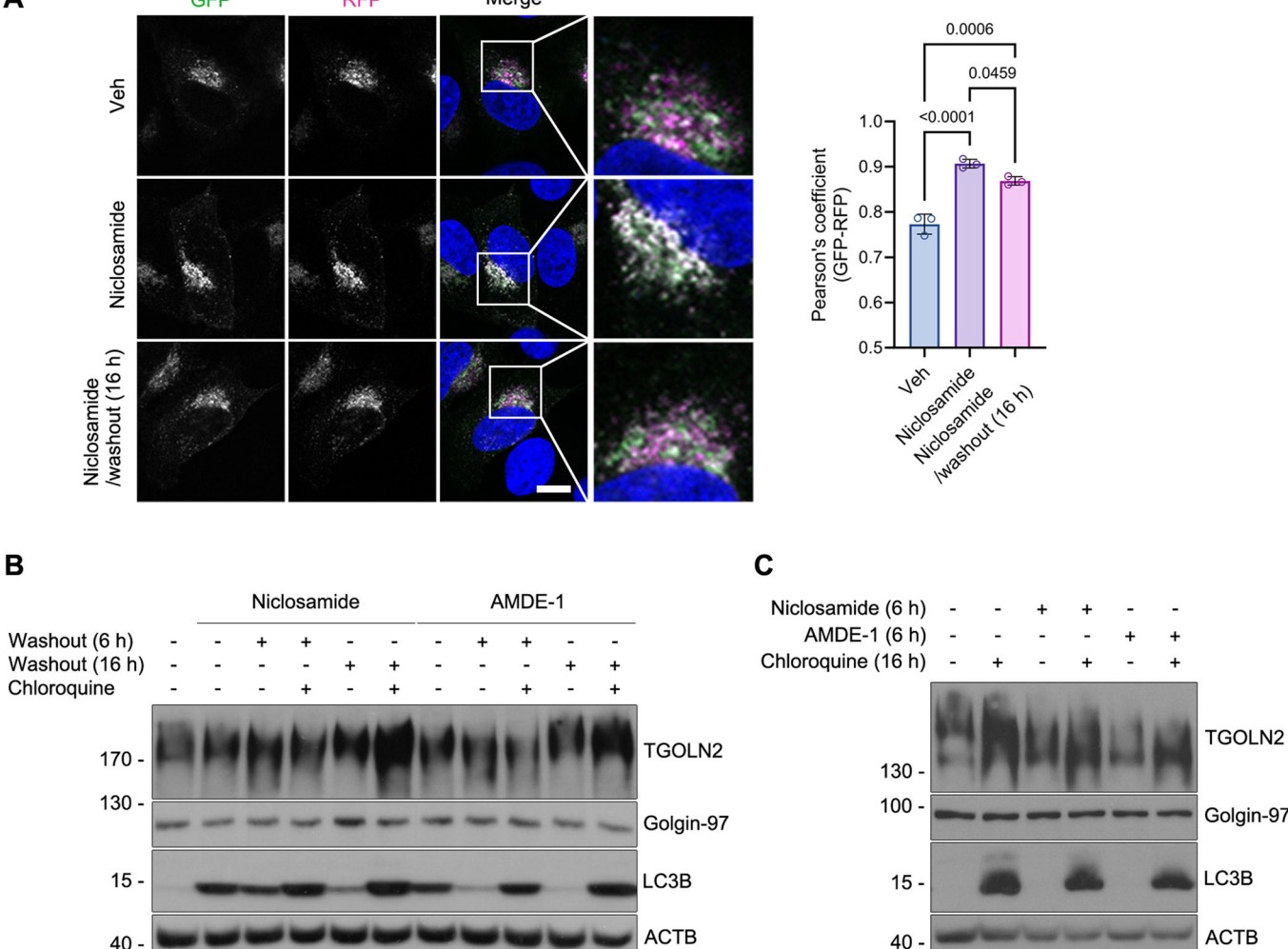

**Figure EV2. LC3-lipidated Golgi membranes are not degraded by lysosomes.**

(**A**) HeLa cells expressing TGOLN2-RFP-GFP were treated with 10 μM niclosamide for 6 h, and niclosamide was then removed for an additional 16 h. Cells were observed by confocal microscopy. Nuclei were stained by Hoechst dye 33342. Scale bar, 10 μm (left). Pearson's correlation coefficient of GFP and RFP is represented as mean ± s.d. ($n = 3$, 33–47 cells per experiment, one-way ANOVA followed by Tukey's multiple comparisons test, Veh vs Niclosamide; $p = 0.000084$) (right). (**B**, **C**) HeLa cells were treated with 10 μM niclosamide or 10 μM AMDE-1 for 6 h and niclosamide and AMDE-1 were removed for an additional 6 h (**B**) or 16 h (**B**, **C**) in the presence or absence of 50 μM chloroquine. Vehicle-treated cells were further incubated with 50 μM chloroquine for 16 h in (**C**). Cells were subjected to immunoblot analysis.

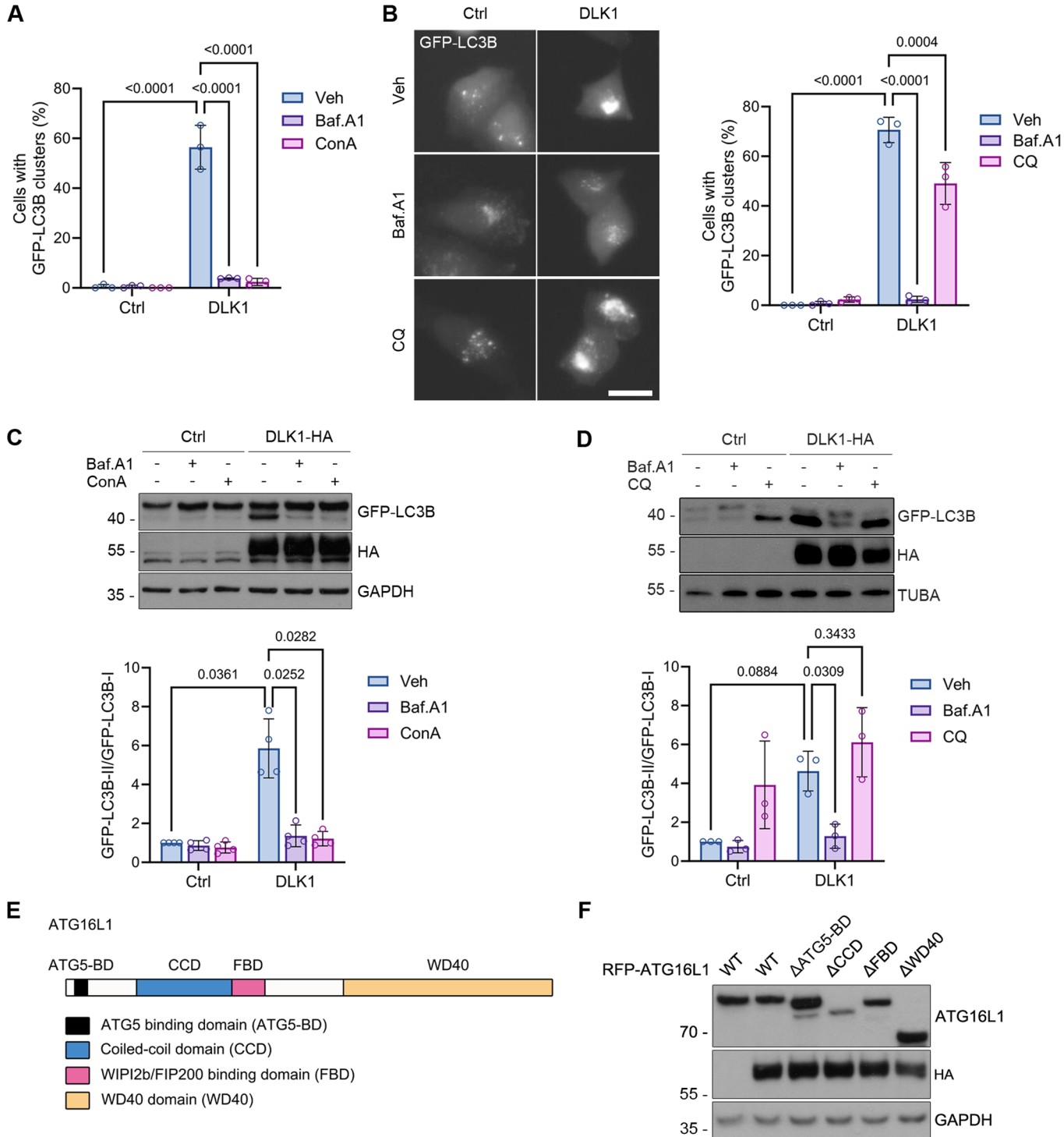

◄ **Figure EV3.  V-ATPase mediates Golgi-LC3 lipidation.**

(**A**) HeLa cells expressing DLK1-HA and GFP-LC3B were treated with 20 nM bafilomycin A1 (Baf.A1) or 200 nM concanamycin A (ConA) for 6 h and observed by fluorescence microscopy. The percentages of cells with GFP-LC3B clusters are represented as mean ± s.d. ($n = 3$, 90–199 cells per experiment, two-way ANOVA followed by Tukey's multiple comparisons test, Ctrl/Veh vs DLK1/Veh; $p = 0.0000000041$, DLK1/Veh vs Baf.A1; $p = 0.0000000079$, DLK1/Veh vs ConA; $p = 0.0000000060$). (**B**) HeLa cells expressing DLK1-HA and GFP-LC3B were treated with 20 nM bafilomycin A1 (Baf.A1) or 100 μM chloroquine (CQ) for 6 h and observed by fluorescence microscopy. Scale bar, 20 μm (left). The percentages of cells with GFP-LC3B clusters are represented as mean ± s.d. ($n = 3$, 139–240 cells per experiment, two-way ANOVA followed by Tukey's multiple comparisons test, Ctrl/Veh vs DLK1/Veh; $p = 0.0000000010$, DLK1/Veh vs Baf.A1; $p = 0.0000000015$) (right). (**C, D**) HeLa cells expressing DLK1-HA and GFP-LC3B were treated with 20 nM bafilomycin A1 (Baf.A1) or either 200 nM concanamycin A (ConA) (**C**) or 100 μM chloroquine (CQ) (**D**) for 6 h and subjected to immunoblot analysis (top). Relative signals of GFP-LC3B-II and GFP-LC3B-I on the blots are represented as mean ± s.d. [$n = 4$ independent experiments (**C**), $n = 3$ independent experiments (**D**), two-way ANOVA followed by Tukey's multiple comparisons test] (bottom). (**E**) Schematic representation of ATG16L1 domains. (**F**) Immunoblot analysis of HeLa sgATG16L1 cells expressing DLK1-HA and RFP-ATG16L1 mutants.

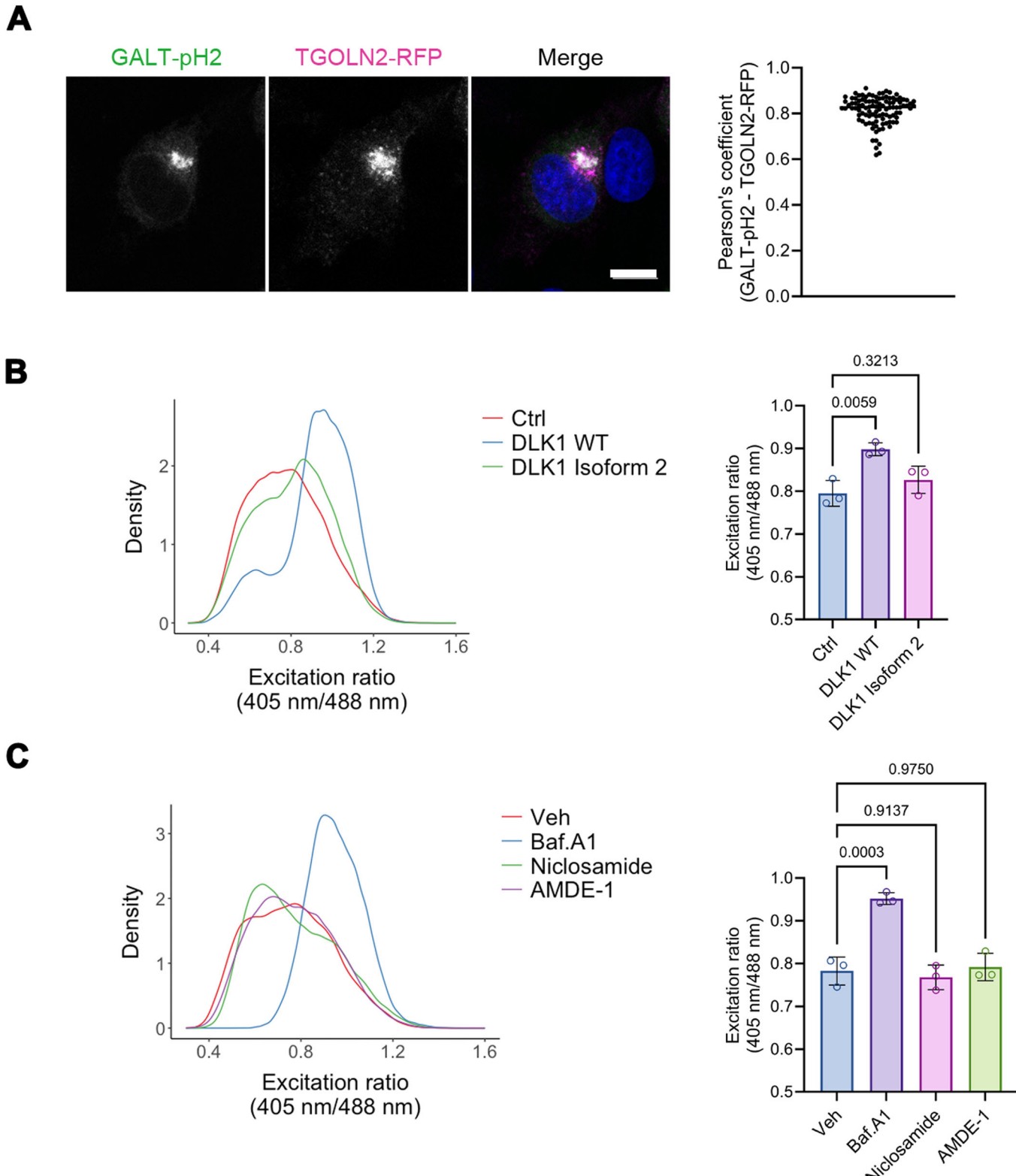

**Figure EV4.  DLK1, but not niclosamide and AMDE-1, increases Golgi pH.**

(**A**) Confocal images of HeLa cells expressing GALT-pH2 and TGOLN2-RFP. Nuclei were stained by Hoechst dye 33342. Scale bar, 10 µm (left). Pearson's correlation coefficient of GALT-pH2 and TGOLN2-RFP ($n = 104$ cells) was quantified (right). (**B**, **C**) HeLa cells expressing GALT-pH2 and either pcDNA3-HA (Ctrl), DLK1-HA WT, or Isoform 2 were subjected to flow cytometry analysis (**B**). HeLa cells expressing GALT-pH2 were treated with 20 nM bafilomycin A1 (Baf.A1), 10 µM niclosamide, or 10 µM AMDE-1 for 6 h and subjected to flow cytometry analysis (**C**). Frequency distributions of excitation ratios (405/408 nm) are shown (left). Median values of the excitation ratios are represented as mean ± s.d. ($n = 3$ independent experiments, one-way ANOVA followed by Tukey's multiple comparisons test) (right).

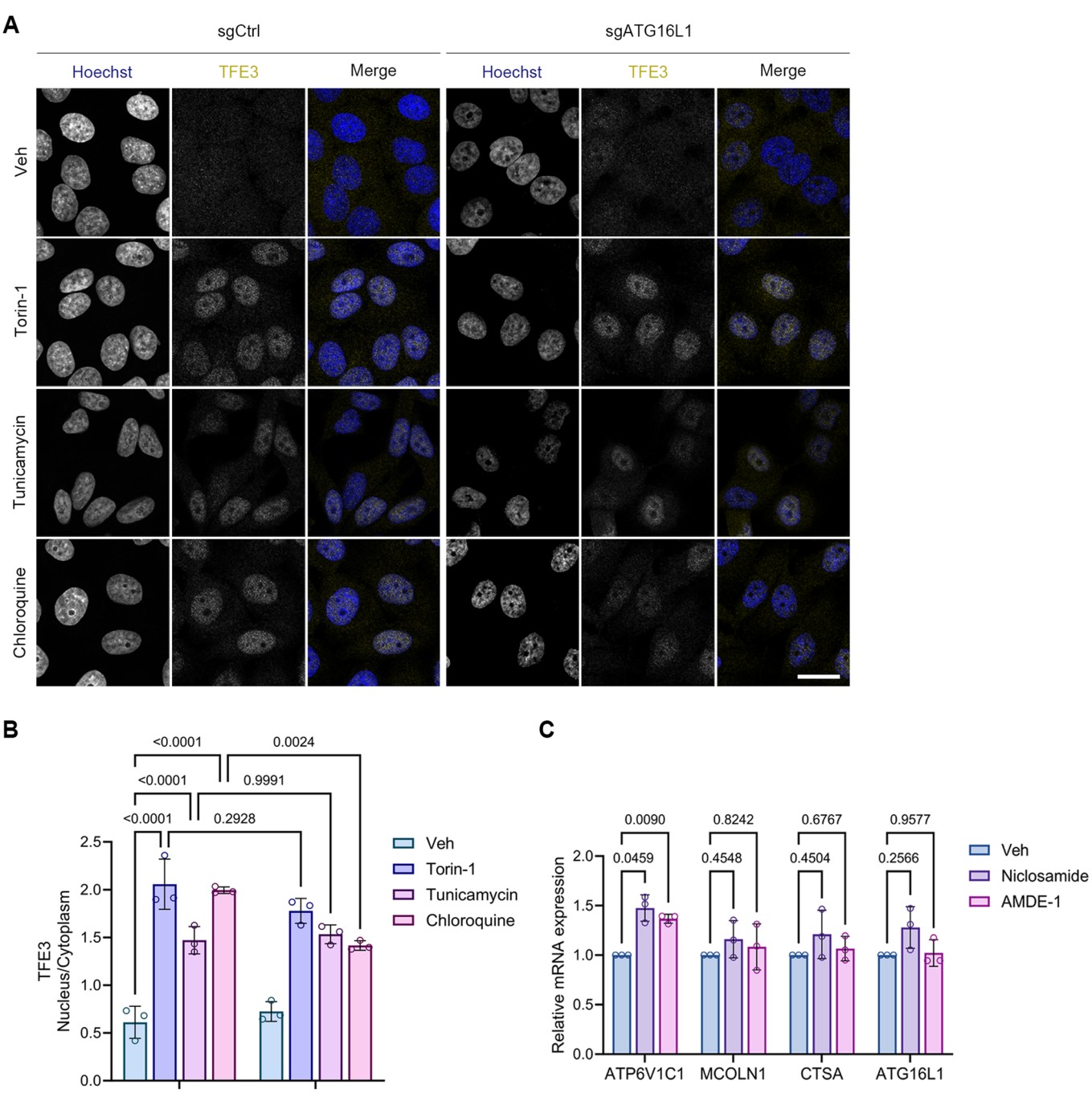

**Figure EV5.  Golgi stress-specific function of LC3 lipidation in TFE3 regulation.**

(A, B) Confocal images of HeLa sgCtrl and sgATG16L1 cells incubated with 250 nM Torin-1 (1 h), 2 mg/ml tunicamycin (16 h), or 50 μM chloroquine (2 h) and immunostained with anti-TFE3 antibody. Nuclei were stained by Hoechst dye 33342. Scale bar, 20 μm (A). The nucleus/cytoplasm ratio of TFE3 fluorescence intensity is represented as mean ± s.d. ($n = 3$, 109–176 cells per experiment, two-way ANOVA followed by Tukey's multiple comparisons test, sgCtrl/Veh vs Torin-1; $p = 0.000000024$, sgCtrl/Veh vs Tunicamycin; $p = 0.000029472$, sgCtrl/Veh vs Chloroquine; $p = 0.000000046$) (B). (C) RNA of HeLa cells exposed to 10 μM niclosamide or 10 μM AMDE-1 for 6 h was analyzed with quantitative real-time PCR. Bars represent mean ± s.d. ($n = 3$ independent experiments, one-way ANOVA followed by Dunnett's multiple comparisons test).

