## [Peer Review File · The EMBO Journal]

Non-autophagic Golgi-LC3 lipidation facilitates TFE3 stress response against Golgi dysfunction

Yong-Keun Jung, Jaemin Kang, Cathena Meiling Li, Namhoon Kim, and Jongyeon Baek

Corresponding author(s): Yong-Keun Jung (ykJung@snu.ac.kr)

Review Timeline:

Submission Date:	8th Feb 24
Editorial Decision:	11th Mar 24
Revision Received:	10th Jul 24
Editorial Decision:	13th Aug 24
Revision Received:	19th Aug 24
Accepted:	21st Aug 24

Editor: William Teale

Transaction Report:

Dear Prof. Jung,

Thank you for submitting your manuscript for consideration by the EMBO Journal. It has now been seen by three referees whose comments are shown below.

As discussed on Zoom, given the referees' positive recommendations, I would like to invite you to submit a revised version of the manuscript, addressing the comments of all three reviewers. I should add that it is EMBO Journal policy to allow only a single round of revision, and acceptance of your manuscript will therefore depend on the completeness of your responses in this revised version.

Thank you for the opportunity to consider your work for publication. I look forward to your revision.

Best regards,

William

William Teale, Ph.D.
Editor
The EMBO Journal

When submitting your revised manuscript, please carefully review the instructions below and include the following items:

- 1) a .docx formatted version of the manuscript text (including legends for main figures, EV figures and tables). Please make sure that the changes are highlighted to be clearly visible.
- 2) individual production quality figure files as .eps, .tif, .jpg (one file per figure).
- 3) a .docx formatted letter INCLUDING the reviewers' reports and your detailed point-by-point response to their comments. As part of the EMBO Press transparent editorial process, the point-by-point response is part of the Review Process File (RPF), which will be published alongside your paper.
- 4) a complete author checklist, which you can download from our author guidelines ([https://wol-prod-cdn.literatumonline.com/pb-assets/embo-site/Author Checklist%20-%20EMBO%20J-1561436015657.xlsx](https://wol-prod-cdn.literatumonline.com/pb-assets/embo-site/Author%20Checklist%20-%20EMBO%20J-1561436015657.xlsx)). Please insert information in the checklist that is also reflected in the manuscript. The completed author checklist will also be part of the RPF.
- 5) Please note that all corresponding authors are required to supply an ORCID ID for their name upon submission of a revised manuscript.
- 6) We require a 'Data Availability' section after the Materials and Methods. Before submitting your revision, primary datasets produced in this study need to be deposited in an appropriate public database, and the accession numbers and database listed under 'Data Availability'. Please remember to provide a reviewer password if the datasets are not yet public (see <https://www.embopress.org/page/journal/14602075/authorguide#datadeposition>). If no data deposition in external databases is needed for this paper, please then state in this section: This study includes no data deposited in external repositories. Note that the Data Availability Section is restricted to new primary data that are part of this study.

Note - All links should resolve to a page where the data can be accessed.

- 7) When assembling figures, please refer to our figure preparation guideline in order to ensure proper formatting and readability in print as well as on screen:
<http://bit.ly/EMBOPressFigurePreparationGuideline>

8) For data quantification: please specify the name of the statistical test used to generate error bars and P values, the number (n) of independent experiments (specify technical or biological replicates) underlying each data point and the test used to calculate p-values in each figure legend. The figure legends should contain a basic description of n, P and the test applied. Graphs must include a description of the bars and the error bars (s.d., s.e.m.).

9) We would also encourage you to include the source data for figure panels that show essential data. Numerical data can be provided as individual .xls or .csv files (including a tab describing the data). For 'blots' or microscopy, uncropped images should be submitted (using a zip archive or a single pdf per main figure if multiple images need to be supplied for one panel). Additional information on source data and instruction on how to label the files are available at .

10) We replaced Supplementary Information with Expanded View (EV) Figures and Tables that are collapsible/expandable online (see examples in <https://www.embopress.org/doi/10.15252/embj.201695874>). A maximum of 5 EV Figures can be typeset. EV Figures should be cited as 'Figure EV1, Figure EV2' etc. in the text and their respective legends should be included in the main text after the legends of regular figures.

12) Our journal encourages inclusion of *data citations in the reference list* to directly cite datasets that were re-used and obtained from public databases. Data citations in the article text are distinct from normal bibliographical citations and should directly link to the database records from which the data can be accessed. In the main text, data citations are formatted as follows: "Data ref: Smith et al, 2001" or "Data ref: NCBI Sequence Read Archive PRJNA342805, 2017". In the Reference list, data citations must be labeled with "[DATASET]". A data reference must provide the database name, accession number/identifiers and a resolvable link to the landing page from which the data can be accessed at the end of the reference. Further instructions are available at .

Further instructions for preparing your revised manuscript:

We realize that it is difficult to revise to a specific deadline. In the interest of protecting the conceptual advance provided by the work, we recommend a revision within 3 months (9th Jun 2024). Please discuss the revision progress ahead of this time with the editor if you require more time to complete the revisions. Use the link below to submit your revision:

Referee #1:

In autophagy, ATG8 proteins are important regulatory proteins, which are conjugated via the ATG12-ATG5-ATG16L1 complex to their substrate lipid phosphatidylethanolamine (PE) resulting in the formation of double-membraned autophagosomes. Besides autophagosomes, lipidated ATG8 proteins can be recruited to several single membrane compartments known as CASM, which consists of distinct molecular features separated from canonical autophagy. Recent studies have shown that several Golgi-damaging agents can induce LC3B lipidation on the Golgi apparatus, however the molecular mechanism is still poorly understood. The authors found that overexpression of DLK1 induced intracellular ATG8 accumulation on the trans-Golgi network. While immunoblotting showed that DLK1 overexpression increased LC3B lipidation, transmission electron microscopy analysis revealed no accumulation of double-membraned autophagosomes. To investigate which specific ATG protein complex was responsible for regulating DLK1-induced LC3B lipidation, FIP200 and ATG16L1 were deleted. Intriguingly, FIP200-KO was unable to inhibit LC3B accumulation, however ATG16L1-KO prevented LC3B clustering as well as the lipidation. Moreover, LC3B-G120A mutant did neither show any clustering at the trans-Golgi network nor LC3B conversion, indicating that the DLK1-induced LC3B lipidation is indeed dependent on the ATG12-ATG5-ATG16L1 complex. To systemically characterize LC3B lipidation at the Golgi apparatus, Golgi damage was induced by a variety of different conditions. Among these AMDE-1 and niclosamide showed robust LC3B clustering on the Golgi. Using the RUSH system, the authors demonstrated that post-Golgi trafficking was inhibited by DLK1 overexpression. Moreover, the authors showed that recruitment of ATG16L1 to the trans-Golgi network depended on DLK1 overexpression and was sensitive to Baf.A1 and concanamycin A treatment. Consistently, the ATG16L1-ATP6V1A interaction was adversatively affected by DLK1 overexpression and BafA1 and lost upon deletion of ATG16L1's WD40 domain. Importantly, the authors found that post-Golgi trafficking inhibition triggered nuclear shuttling of the transcription factor TFE3 and transcription of Golgi stress response genes. Overall, the authors provide a comprehensive analysis of the role of LC3 lipidation to the trans-Golgi in response to post-Golgi trafficking inhibition. In general, the first part (Figure 1-4) is cohesive and convincing while the second part (Figure 5 and 6) is rather weak (please see points below).

Major points

- 1) The authors should show that ATP6V1A colocalize with ATG16L1 at the trans-Golgi upon DLK1-HA expression.
- 2) The authors need to show the expression levels of the different ATG16L1 constructs as control for Figure 4G. Moreover, it seems that the ATG16L1 lacking the WD40 domain still colocalizes with TGOLN2. Maybe quantification can help to substantiate the authors' claim that "loss of the WD40 domain abolished colocalization of ATG16L1 and TGOLN2".
- 3) In Figure 5F, loss of ATG16L1 does not seem to quantitatively affect translocation of TFE3. In fact, in ATG16L1 KO cells the majority of TFE3 is actually in the nucleus. How does this fit to the proposed LC3 lipidation dependent translocation model?
- 4) The authors show that ATG16L1 KO inhibits the nuclear translocation of TFE3 in response to post-Golgi trafficking stress. So why did the expression pattern of TFE3 target genes not change in ATG16L1 KO cells after treatment with the Golgi-damaging reagent, when TFE3 is no longer able to translocate to the nucleus and act as a transcription factor?
- 4) Did the authors also check the expression of TFE3 target genes in response to DLK1 overexpression as they did for niclosamide and AMDE-1? Especially since the authors state that DLK1 overexpression is able to possibly exclude non-specific effects of the chemical treatments?
- 5) The pathophysiological relevance of the trans-Golgi LC3 lipidation is not entirely clear. Are there any diseases related to post-Golgi trafficking inhibition? Could LC3 lipidation be tested in these conditions? At least the authors should discuss this aspect.

Minor points

- 6) Figure EV7: For the proximity labelling experiment TurboID-LC3B transfected cells were biotinylated for 6 hours with 100 μ M. Why didn't the authors use 10 min biotinylation with 50 μ M biotin as recommend in the TurboID protocol?
- 7) Figure EV7: Why do in the Western Blot after the streptavidin pulldown and staining with anti-TFE3 bands appear around the

size of 45 kDa? It would be clearer if the authors would show the TFE3 Input blot from 100 kDa to 45 kDa.

8) Figure 5 (A): There is a small typo in the headline "sgATG16L1 #3", since the authors state in the figure legend that they used sgATG16L1 #2.

9) Figure 5 (A): What is the difference between sgATG16L1 #1 and sgATG16L1 #2?

10) Figure 6 (C): It seems like the image for calcein-AM in the vehicle control is overexposed compared to the Golgi-damaging reagents.

Referee #2:

This paper by Kang et al entitled "Non-autophagic Golgi-LC3 lipidation facilitates TFE3 stress response against Golgi dysfunction" presents data to suggest that Golgi-LC3 lipidation via the CASM (conjugation of ATG8s to single membranes) pathway occurs because of post-Golgi trafficking inhibition and leads to nuclear translocation of TFE3 to alleviate damage to the Golgi apparatus.

This is an interesting study adding more pieces to the puzzle depicting the roles of conjugation of ATG8/LC3 to the Golgi membranes. The experiments performed are very well executed and the data presented are generally convincing and quantified where this is required. I also found the Discussion section interesting to read.

However, a number of studies have recently been published describing CASM occurring in a number of stress processes at several organelles, most particularly at damaged lysosomes and important mechanistic findings have already been published. There have also been several previous reports on Golgi-damaging conditions leading to CASM at Golgi membranes. Hence, for this study to move from an important, but incremental study, to clearly stand out as bringing important new mechanistic insight of the role of CASM at the Golgi following post-Golgi transport inhibition some effort should be done by the authors to fill some of the holes in our present understanding.

Major comments:

1) How is the post Golgi trafficking inhibition sensed and how is this conveyed to the conjugation machinery to mediate the LC3 lipidation at the Golgi membranes?

The authors write that "Golgi-LC3 lipidation also can be coupled to ion and pH imbalances in Golgi apparatus. A plausible model is that post-Golgi trafficking inhibition affects ion and pH imbalances in Golgi lumen, consequently leading Golgi-LC3 lipidation."

I think this is a very good suggestion and the authors should do experiments to test this model/hypothesis in a revised version.

2) How does LC3-single membrane lipidation to Golgi membranes signal to mediate nuclear translocation of TFE3?

Here the authors write in the Discussion : "As TFE3 does not directly bind to LC3 (Fig. EV7), Golgi-LC3 lipidation may activate Golgi stress-specific phosphatases to dephosphorylate TFE3 for its nuclear translocation. Considering TFE3 dephosphorylation by calcineurin under ER stress (Martina et al., 2016) and calcium efflux by LC3 lipidation under lysosomal stress (Nakamura et al., 2020), LC3 lipidation may also be involved in calcium efflux from the Golgi apparatus, leading to TFE3 dephosphorylation via calcineurin."

Indeed, a natural next step would be to test this

3) What is the destiny or fate of the Golgi membranes that are LC3-lipidated? Are they degraded by autophagy? It did not look from the images that the Golgi became fragmented.

Referee #3:

Although recent studies have shown that ATG8/LC3 is lipidated with PS in the Golgi (CASM) upon Golgi stress, its molecular mechanism has been unknown. In this manuscript, the authors found that overexpression of DLK1 induces Golgi-LC3 lipidation via ATG12-ATG5-ATG16L1 complex, resulting in nuclear translocation of TFE3 (a transcription factor regulating the Golgi stress response). In addition, they revealed that inhibition of post-Golgi trafficking precedes Golgi-LC3 lipidation. Interestingly, Golgi-LC3 lipidation is cytoprotective against Golgi stress via transcriptional activation of Golgi-related genes by TFE3. From these observations, the authors concluded that non-autophagic Golgi-LC3 lipidation activates the TFE3 pathway of the Golgi stress response to cope with Golgi dysfunction.

The reviewer thinks that their data are ample and clear to support their conclusion, and that the manuscript would become suitable for publication in the EMBO Journal after minor revision.

(1) How does overexpression of DLK1 cause Golgi stress and Golgi-LC3 lipidation? What is the known function of DLK1?

(2) Does overexpression of DLK1 inhibit post-Golgi protein trafficking? How about overexpression of other secretory proteins? The reviewer wonder if overexpression of secretory proteins such as DLK1 overwhelms Golgi function, which causes Golgi stress (inhibition of post-Golgi protein trafficking) and finally TFE3 activation.

(3) It would be better to describe authors speculation on molecular mechanism how Golgi-lipidation of LC3 induces dephosphorylation of TFE3.

Our response to the comments of Reviewer #1

In autophagy, ATG8 proteins are important regulatory proteins, which are conjugated via the ATG12-ATG5-ATG16L1 complex to their substrate lipid phosphatidylethanolamine (PE) resulting in the formation of double-membraned autophagosomes. Besides autophagosomes, lipidated ATG8 proteins can be recruited to several single membrane compartments known as CASM, which consists of distinct molecular features separated from canonical autophagy. Recent studies have shown that several Golgi-damaging agents can induce LC3B lipidation on the Golgi apparatus, however the molecular mechanism is still poorly understood. The authors found that overexpression of DLK1 induced intracellular ATG8 accumulation on the trans-Golgi network. While immunoblotting showed that DLK1 overexpression increased LC3B lipidation, transmission electron microscopy analysis revealed no accumulation of double-membraned autophagosomes. To investigate which specific ATG protein complex was responsible for regulating DLK1-induced LC3B lipidation, FIP200 and ATG16L1 were deleted. Intriguingly, FIP200-KO was unable to inhibit LC3B accumulation, however ATG16L1-KO prevented LC3B clustering as well as the lipidation. Moreover, LC3B-G120A mutant did neither show any clustering at the trans-Golgi network nor LC3B conversion, indicating that the DLK1-induced LC3B lipidation is indeed dependent on the ATG12-ATG5-ATG16L1 complex. To systemically characterize LC3B lipidation at the Golgi apparatus, Golgi damage was induced by a variety of different conditions. Among these AMDE-1 and niclosamide showed robust LC3B clustering on the Golgi. Using the RUSH system, the authors demonstrated that post-Golgi trafficking was inhibited by DLK1 overexpression. Moreover, the authors showed that recruitment of ATG16L1 to the trans-Golgi network depended on DLK1 overexpression and was sensitive to Baf.A1 and concanamycin A treatment. Consistently, the ATG16L1-ATP6V1A interaction was adversely affected by DLK1 overexpression and BafA1 and lost upon deletion of ATG16L1's WD40 domain. Importantly, the authors found that post-Golgi trafficking inhibition triggered nuclear shuttling of the transcription factor TFE3 and transcription of Golgi stress response genes. Overall, the authors provide a comprehensive analysis of the role of LC3 lipidation to the trans-Golgi in response to post-Golgi trafficking inhibition. In general, the first part (Figure 1-4) is cohesive and convincing while the second part (Figure 5 and 6) is rather weak (please see points below).

Major points

Q1) The authors should show that ATP6V1A colocalize with ATG16L1 at the trans-Golgi upon DLK1-HA expression.

Responses: Following the reviewer's recommendation, we performed additional experiments to observe localization of endogenous ATP6V1A with ATG16L1 and TGOLN2 with immunostaining assays. We first did the experiment using anti-ATP6V1A antibody (Proteintech, 17115-1-AP) which was previously utilized for western blot analysis in Fig. 4C-F. Unfortunately, immunoreactivity of the ATP6V1A signal was observed largely in the nucleus but not in the V-ATPase-positive vesicles of the cytoplasm in HeLa cells (**Reviewer Figure. 1A,B**), suggesting that this antibody from Proteintech is inappropriate for the immunocytochemistry of ATP6V1A. We then tried the same experiments to observe V1 domain of V-ATPase with other antibodies, anti-ATP6V1A antibody (ABclonal, A14706), anti-ATP6V1B1 antibody (ABclonal, A6876), and anti-ATP6V1B2 antibody (ABclonal, A3754). Even with the use of these three additional antibodies, we were unable to observe the proper cytoplasmic pattern of V1 domain of V-ATPase (data not shown). We have also tried to image V1 domain of V-ATPase in HeLa cells after ectopic expression of GFP-tagged V1 subunits, GFP-ATP6V1A and GFP-ATP6V1B2. With unknown reason, however, we could

also not observe proper subcellular localization of V-ATPase V1 domain. We reasoned that the inability to observe V1 subunit of V-ATPase with immunostaining might be due to limitations

Reviewer Fig.1

for the antibody to access to V1 subunit after the formation of V-ATPase holocomplex. While we have done our best to address the point, we have a limitation in the antibody available for the immunostaining of ATP6V1A. We would appreciate if the reviewer is able to provide a generous suggestion for a suitable antibody or alternative imaging methods. We would then re-address the point. We apologize for our inability to produce satisfactory results and appreciate reviewer's understanding.

(Reviewer Fig. 1) ATP6V1A localization upon DLK1 overexpression.

(A, B) Confocal images of ATP6V1A in HeLa cells expressing DLK1-HA and TGOLN2-GFP (A) or RFP-ATG16L1 (B). Cells were immunostained with anti-ATP6V1A antibody and nuclei were stained by Hoechst dye 33342. Scale bars, 10 μ m.

Q2) The authors need to show the expression levels of the different ATG16L1 constructs as control for Figure 4G. Moreover, it seems that the ATG16L1 lacking the WD40 domain still colocalizes with TGOLN2. Maybe quantification can help to substantiate the authors' claim that "loss of the WD40 domain abolished colocalization of ATG16L1 and TGOLN2".

Responses: We agree with reviewer that expression levels of different ATG16L1 constructs should be examined and quantification of the colocalization between ATG16L1 and TGOLN2 is necessary. Because ATG16L1 dimerizes mainly via its coiled-coil domain (CCD) (Gammoh, 2020), we concerned that exogenous RFP-ATG16L1 might dimerize with endogenous ATG16L1, which might interfere with our understanding on subcellular localization of ATG16L1 mutants. We thus expressed ATG16L1 mutants in HeLa ATG16L1 KO cells and confirmed their expression level and subcellular localization under DLK1 overexpression. Except for RFP-ATG16L1 CCD, expression levels were not significantly different among ATG16L1 mutants we have tested (**New Fig. EV3F**). We have added these results into **Fig. EV3F**.

New Fig. EV3F

(New Fig. EV3F) Protein expression of ATG16L1 mutants.

Immunoblot analysis of HeLa sgATG16L1 cells expressing DLK1-HA and RFP-ATG16L1 mutant.

Following the suggestion, we also quantified confocal microscopic images obtained from the repeated experiments with Fiji program and could now better demonstrate the colocalization pattern between ATG16L1 mutants and TGOLN2 with Pearson's coefficient and statistical significance (**New Fig. 4G**). With these assays, we better show the role of WD40 domain in

ATG16L1 recruitment to the Golgi apparatus under DLK1 overexpression. We replaced the previous Figure with these. (236 - 240 lines on page 11).

New Fig. 4G

(New Fig. 4G) Colocalization of ATG16L1 mutants with TGOLN2 under DLK1 overexpression.

Confocal images of HeLa sgATG16L1 cells expressing DLK1-HA, TGOLN2-GFP and RFP-ATG16L1 mutants. Nuclei were stained by Hoechst dye 33342. Scale bar, 10 μ m (left). Colocalization of TGOLN2-GFP and RFP-ATG16L1 mutants was quantified with Fiji and represented as Pearson's correlation coefficient. Bars represent mean \pm s.d. ($n = 4, 21 \sim 67$ cells per experiment, one-way ANOVA followed by Tukey's multiple comparisons test) (right).

Q3 In Figure 5F, loss of ATG16L1 does not seem to quantitatively affect translocation of TFE3. In fact, in ATG16L1 KO cells the majority of TFE3 is actually in the nucleus. How does this fit to the proposed LC3 lipidation dependent translocation model?

Responses: We entirely agree with reviewer's concern about Fig. 5F in which TFE3 protein expression is elevated in both nucleus fraction and whole cell lysates. First, we would like to explain about methodological limitation of this experiment. Because the nucleus fractions were prepared from total cells after DLK1 transfection, some cells among total cells were not transfected. It thus contains proteins from non-transfected cells. Second, in our experimental conditions using polymer-based transfection reagents, transfection efficiency (the percentages of transfected cells) was lower in ATG16L1 KO cells than control cells. Unlike the polymer reagents, transfection with lipofectamine 2000 (Thermo Fisher Scientific) allowed similar levels of DLK1 expression between ATG16L1 WT and KO cells. On other hand, this experimental condition led to some increase of TFE3 protein levels in both nucleus fraction and whole cell lysates. When we examined the proportion of TFE3 in the nucleus only under DLK1 overexpression condition, no difference was observed between ATG16L1 WT and KO cells (**Previous Fig. 5H**).

New Fig. 5H

overexpression condition relative to the control condition, however, loss of ATG16L1 attenuated DLK1-induced nuclear translocation of TFE3 (**New Fig. 5H**). To clarify this point, we performed two additional experiments and present those results together with the statistical significance for the difference.

(New Fig. 5H) ATG16L1 KO inhibits DLK1-induced TFE3 nuclear translocation.

Relative signal intensities of nuclear/cytoplasmic TFE3 between non-transfected and DLK1-transfected are represent as mean \pm s.d. ($n = 5$, two-tailed Student's t test).

Q4) The authors show that ATG16L1 KO inhibits the nuclear translocation of TFE3 in response to post-Golgi trafficking stress. So why did the expression pattern of TFE3 target genes not change in ATG16L1 KO cells after treatment with the Golgi-damaging reagent, when TFE3 is no longer able to translocate to the nucleus and act as a transcription factor?

Responses: First, we apologize for lacking detailed explanation on Previous Fig. 6B (New Fig. 6C) in which expression of TFE3 target genes was measured in ATG16L1 KO cells. Among four TFE3 target genes (SIAT4A, GCP60, WIPI49, and STX3A), niclosamide-induced mRNA expression of WIPI49 and STX3A, which are related to vesicle transport (Taniguchi *et al.*, 2015), was significantly inhibited by ATG16L1 knockout. Compared to WIPI49 and STX3A, GCP60 expression was much less reduced but showed a statistically significant difference in ATG16L1 KO cells. On the other hand, SIAT4A expression was not significantly affected by ATG16L1 knockout although it showed a slight decrease. The previous results in Fig. 5A and C indicate that loss of ATG16L1 decreases, but not completely, TFE3 nuclear translocation under Golgi stress. Therefore, we believe that expression of TFE3 target genes is not likely to be completely suppressed, resulting in distinct expression patterns and regulation of TFE3 target genes. In case of MCL1-S, a target gene of the MAPK-ETS pathway, its upregulation by Golgi-damaging reagents is not affected by ATG16L1 knockout, suggesting the presence of LC3-independent regulation of this pathway. We added detailed explanations in result section of the manuscript. (335 - 340 lines on page 15)

Q5) Did the authors also check the expression of TFE3 target genes in response to DLK1 overexpression as they did for niclosamide and AMDE-1? Especially since the authors state that DLK1 overexpression is able to possibly exclude non-specific effects of the chemical treatments?

Responses: We fully agree with reviewer's comments and performed additional experiments to examine the effects of DLK1 overexpression on mRNA levels of TFE3 target genes as well. Quantitative real-time PCR analysis showed that DLK1 overexpression significantly increased expression of TFE3 target genes, including SIAT4A, GCP60, Giantin, WIPI49, and STX3A, in HeLa cells (New Fig. 6B). In addition, expression of target genes in the HSP47 pathway (HSP47), CREB3 pathway (ARF4), and MAPK-ETS pathway (MCL1-S) was also upregulated by DLK1 overexpression, although statistical significance was not observed in HSP47 and MCL1-S. We added these into new figure. (326 - 327 lines on page 15). Thank you very much.

New Fig. 6B

(New Fig. 6B) The mRNA levels of TFE3 target genes under DLK1 overexpression.

Total RNA of HeLa cells expressing pcDNA3-HA (Ctrl) or DLK1-HA was analyzed with quantitative real-time PCR. Bars represent mean \pm s.d. ($n = 3$, two-way ANOVA followed by Tukey's multiple comparisons test).

Q6) The pathophysiological relevance of the trans-Golgi LC3 lipidation is not entirely clear. Are there any diseases related to post-Golgi trafficking inhibition? Could LC3 lipidation be tested in these conditions? At least the authors should discuss this aspect.

Responses: We thank reviewer for raising this important issue and agree with the necessity for studies on the pathophysiological relevance of Golgi-LC3 lipidation. Definitely, we have tried to examine a role of LC3 lipidation in relevant disease model using mutation of FYVE, RhoGEF and PH domain containing 1 (FGD1). Egorov *et al.* reported that expression of

dominant-negative FGD1 mutant (FGD1-AS) inhibits post-Golgi transport (Egorov *et al*, 2009). In our trial, however, we could not observe its effect on post-Golgi transport inhibition and Golgi-LC3 lipidation in HeLa cells under FGD1-AS overexpression (data not shown). Therefore, it seems that another genetic models should be examined in future studies. As recommended, we added more discussion on the genetic mutations linked to defects in post-Golgi membrane trafficking and the importance of establishing those genetic disorder models to elucidate pathophysiological role of Golgi-LC3 lipidation and TFE3 (426 - 435 lines on page 19). Thank you very much for such valuable suggestion.

Minor points

Q7) Figure EV7: For the proximity labelling experiment TurboID-LC3B transfected cells were biotinylated for 6 hours with 100 μ M. Why didn't the authors use 10 min biotinylation with 50 μ M biotin as recommend in the TurboID protocol?

Responses: We first hypothesized that direct interaction between LC3 and TFE3 could explain the mechanism by which TFE3 dephosphorylation is affected by LC3. To test this hypothesis, we conducted the proximity labeling experiment using TurboID-LC3 under Golgi stress. If TFE3 does indeed bind to LC3, it was unclear at which time point this interaction would occur, making it difficult to set a specific time point for biotin treatment within 10 min. Therefore, we treated cells with biotin for 6 h together with niclosamide, AMDE-1 or DLK1 overexpression. Despite incubation with biotin for a longer time period and at a higher concentration than the recommended conditions, TFE3 was not biotinylated by TurboID-LC3B in our experiments, excluding a possibility for the direct interaction between TFE3 and LC3.

Q8) Figure EV7: Why do in the Western Blot after the streptavidin pulldown and staining with anti-TFE3 bands appear around the size of 45 kDa? It would be clearer if the authors would show the TFE3 Input blot from 100 kDa to 45 kDa.

Previous Fig EV7

Responses: As recommended, we added TFE3 input blots with our repeated experiments, showing bands ranging from 45 kDa to 100 kDa (**New Appendix Fig. S4**). No bands were observed around 45 kDa in both streptavidin pulldown blots and input blots. In the streptavidin pulldown blot of the previous Fig EV7, bands near 45 kDa were observed even in conditions without biotin treatment. We believe these bands are non-specific, TFE3-unrelated bands arisen during the process of protein elution from streptavidin agarose beads in experiments.

New Appendix Fig S4

(**New Appendix Fig. S4**) Previous Fig. EV7 (top) and New Appendix Fig. S4 (bottom).

Q9) Figure 5 (A): There is a small typo in the headline "sgATG16L1 #3", since the authors state in the figure legend that they used sgATG16L1 #2.

Responses: We appreciate reviewer for identifying this typo and corrected it.

Q10) Figure 5 (A): What is the difference between sgATG16L1 #1 and sgATG16L1 #2?

Responses: We are sorry not sufficiently describing it. We generated two ATG16L1 knockout cell lines with CRISPR-Cas9 system by utilizing two independent single guide RNAs to eliminate the possibility of off-target effects. We added more description in materials and methods (478 line on page 21 - 484 line on page 22). Nucleotide sequence of each single guide RNA is described in 'Reagent table' file.

Q11) Figure 6 (C): It seems like the image for calcein-AM in the vehicle control is overexposed compared to the Golgi-damaging reagents.

Responses: As concerned by the reviewer, fluorescence signal of calcein-AM is higher in the vehicle compared to niclosamide/AMDE-1. In all experimental conditions, we have imaged cells under the same gain and exposure time setting. Since Calcein-AM is converted by intracellular esterases into a fluorescent calcein (Uggeri *et al*, 2004), we speculate that 24 h incubation of cells with the Golgi-damaging reagents affected intracellular esterases activity, leading to decreased fluorescence signal compared to control cells.

We appreciate reviewer for valuable comments that strengthen this manuscript.

Our response to the comments of Reviewer #2

This paper by Kang et al entitled "Non-autophagic Golgi-LC3 lipidation facilitates TFE3 stress response against Golgi dysfunction" presents data to suggest that Golgi-LC3 lipidation via the CASM (conjugation of ATG8s to single membranes) pathway occurs because of post-Golgi trafficking inhibition and leads to nuclear translocation of TFE3 to alleviate damage to the Golgi apparatus. This is an interesting study adding more pieces to the puzzle depicting the roles of conjugation of ATG8/LC3 to the Golgi membranes. The experiments performed are very well executed and the data presented are generally convincing and quantified where this is required. I also found the Discussion section interesting to read. However, a number of studies have recently been published describing CASM occurring in a number of stress processes at several organelles, most particularly at damaged lysosomes and important mechanistic findings have already been published. There have also been several previous reports on Golgi-damaging conditions leading to CASM at Golgi membranes. Hence, for this study to move from an important, but incremental study, to clearly stand out as bringing important new mechanistic insight of the role of CASM at the Golgi following post-Golgi transport inhibition some effort should be done by the authors to fill some of the holes in our present understanding.

Major comments:

Q1) How is the post Golgi trafficking inhibition sensed and how is this conveyed to the conjugation machinery to mediate the LC3 lipidation at the Golgi membranes? The authors write that "Golgi-LC3 lipidation also can be coupled to ion and pH imbalances in Golgi apparatus. A plausible model is that post-Golgi trafficking inhibition affects ion and pH imbalances in Golgi lumen, consequently leading Golgi-LC3 lipidation." I think this is a very good suggestion and the authors should do experiments to test this model/hypothesis in a revised version.

Responses: Thanks you for reminding this important issue. In CASM, as mentioned, ion/pH imbalances in single membranes have been suggested as unifying mechanisms as they increase the association of V0 and V1 domains of V-ATPase, resulting in the recruitment of ATG16L1 to the single membranes. To address whether Golgi-LC3 lipidation is coupled to pH imbalances in the Golgi apparatus, we decided to generate a pHluorin-based Golgi apparatus pH sensor. The pHluorin2 is a GFP2 variant which displays bimodal excitation spectrum peaks at 395 nm and 475 nm and an emission maximum at 509 nm (Mahon, 2011). Acidification decreases pHluorin2 excitation at 395 nm but increases excitation at 475 nm, leading to decrease of 395/475 nm excitation ratios. We have generated Golgi-pH sensor (GALT-pH2) through conjugation of pHluorin2 to 1-82 amino acids of beta-1,4-galactosyltransferase 1 (B4GALT1), a Golgi-resident membrane protein to locate pHluorin2 in the Golgi lumen. As expected, we confirmed Golgi apparatus localization of GALT-pH2 as it colocalized with TGOLN2-RFP in HeLa cells (**New Fig. EV4A**).

Using GALT-pH2 and flow cytometry analysis, we measured Golgi pH under post-Golgi trafficking inhibitory conditions triggered by DLK1 overexpression or niclosamide or AMDE-1 treatment. Overexpression of DLK1 increased 405/488 nm excitation ratios of GALT-pH2, while DLK1 Isoform 2, which cannot induce Golgi-LC3 lipidation, did not affect 405/488 nm excitation ratios (**New Fig. EV4B**), indicating that DLK1 overexpression increases luminal pH of the Golgi apparatus. In addition, treatment with bafilomycin A1, an inhibitor of V-ATPase, increased Golgi pH but Golgi-damaging reagents, niclosamide and AMDE-1, did not affect Golgi pH (**New Fig. EV4C**). Considering that niclosamide and AMDE-1 induce V-ATPase-ATG16L1 interaction without pH disturbance, these results suggest that in the Golgi apparatus,

unlike other single-membraned structures, pH imbalance is not likely a common prerequisite for the V-ATPase-ATG16L1 axis. We added these results into supplementary information. (247 line on page 11 - 261 line on page 12)

We propose other possible mechanisms by which post-Golgi trafficking inhibition is sensed and conveyed to the V-ATPase-ATG16L1 axis. Phosphatidylinositol 4-phosphate (PI(4)P), a phospholipid highly enriched in the *trans*-Golgi network (De Matteis *et al*, 2013), interacts with V0 A subunit to locate V-ATPase in Golgi apparatus in yeast (Banerjee & Kane, 2017). One preprint also showed that human V0 A subunit could interact with PI(4)P (Mitra & Kane, 2023). Thus, post-Golgi trafficking inhibition might disturb PI(4)P distribution outside the Golgi apparatus and affect V-ATPase V0-V1 assembly at the Golgi apparatus, leading to Golgi-LC3 lipidation via ATG16L1 recruitment. In addition, actin filaments interact with V1 subunits B2 and C1 to maintain V0-V1 assembly at Golgi apparatus (Serra-Peinado *et al*, 2016). Thus, another plausible model is that post-Golgi trafficking inhibition enhances actin filament-mediated V0-V1 assembly at Golgi apparatus, leading to Golgi-LC3 lipidation. We also added these hypotheses in discussion section. (399 - 408 lines on page 18)

New Fig. EV4

(New Fig. EV4) Measurement of Golgi apparatus pH with GALT-pH2 under post-Golgi trafficking inhibition.

(A) Confocal images of HeLa cells expressing GALT-pH2 and TGOLN2-RFP. Nuclei were stained by Hoechst dye 33342. Scale bar, 10 μ m (left). Colocalization of GALT-pH2 and TGOLN2-RFP was quantified as Pearson's correlation coefficient of individual cells (right).

(B, C) HeLa cells expressing pcDNA3-HA (Ctrl), DLK1-HA WT or DLK1 Isoform 2 together with GALT-pH2 were subjected to flow cytometry analysis (B). HeLa cells expressing GALT-pH2 were treated with 20 nM bafilomycin A1 (Baf.A1), 10 μ M niclosamide or 10 μ M AMDE-1 for 6 h and subjected to flow cytometry analysis (C). Frequency distributions of excitation ratios (405/488 nm) are shown (left). Median values of the excitation ratios are represented as mean \pm s.d. ($n = 3$, one-way ANOVA followed by Tukey's multiple comparisons test) (right).

Q2) How does LC3-single membrane lipidation to Golgi membranes signal to mediate nuclear translocation of TFE3? Here the authors write in the Discussion : " As TFE3 does not directly bind to LC3 (Fig. EV7), Golgi-LC3 lipidation may activate Golgi stress-specific phosphatases to dephosphorylate TFE3 for its nuclear translocation. Considering TFE3 dephosphorylation by calcineurin under ER stress (Martina *et al.*, 2016) and calcium efflux by LC3 lipidation under lysosomal stress (Nakamura *et al.*, 2020), LC3 lipidation may also be involved in calcium efflux from the Golgi apparatus, leading to TFE3 dephosphorylation via calcineurin." Indeed, a natural next step would be to test this.

Responses: As discussed in this manuscript and requested by the reviewer, how the lipidated LC3 regulates TFE3 activation is a remaining and important question. Following reviewer's suggestion, we have tried to address this issue by performing additional experiments.

First, we investigated whether calcium is released from the Golgi apparatus under Golgi stress and regulates nuclear translocation of TFE3. With Fluo-4 AM and flow cytometry analysis, we measured cytosolic calcium levels under post-Golgi trafficking inhibition conditions triggered by DLK1 overexpression and niclosamide or AMDE-1 treatment. We found that only niclosamide, but not DLK1 overexpression and AMDE-1, slightly increased cytosolic calcium levels in HeLa cells (**New Appendix Fig. S5A,B**), suggesting that calcium efflux might not be commonly induced by post-Golgi trafficking inhibition. In addition, we compared calcium release in ATG16L1 WT and KO cells to examine whether Golgi-LC3 lipidation similarly induces calcium release as it does in lysosomes. Compared to WT cells, a relatively low level of cytosolic calcium was observed in ATG16L1 KO cells (**New Appendix Fig. S5A,B**). In contrast to lysosomes, defects in LC3 lipidation rather increases cytosolic calcium levels under Golgi stress, suggesting that Golgi-LC3 lipidation does not induce calcium efflux into the cytosol. Consistent with these results, incubation with calcium chelator BAPTA-AM did not inhibit TFE3 nuclear translocation induced by niclosamide and AMDE-1 treatment (**New**

Appendix Fig. S5

Appendix Fig. S5C). These data suggest that TFE3 activation under post-Golgi trafficking inhibition is not related to calcium release from the Golgi apparatus.

(New Appendix Figure S5) Calcium-independent TFE3 regulation by LC3 lipidation under Golgi stress.

(A, B) HeLa sgCtrl and sgATG16L1 cells exposed to 10 μ M niclosamide or 10 μ M AMDE-1 for 6 h (A) or HeLa sgCtrl and sgATG16L1 cells expressing pcDNA3-HA (Ctrl) or DLK1-HA (B) were treated with 1 μ M Fluo-4 AM and subjected to flow cytometry analysis. Median values of the excitation ratios are represented as mean \pm s.d. ($n = 3$, two-way ANOVA followed by Tukey's multiple comparisons test).

(C) HeLa cells were treated with 10 μ M niclosamide or 10 μ M AMDE-1 for 6 h in the presence or absence of 10 μ M BAPTA-AM. Cells were immunostained with anti-TFE3 antibody and observed by confocal microscopy. Nuclei were stained by Hoechst dye 33342. Scale bar, 20 μ m.

As suggested, further research on Golgi stress sensor and kinase and phosphatase of TFE3 pathway will provide insight how LC3 lipidation regulates TFE3. Accordingly, we have revised discussion section with this suggestion. (420 - 425 lines on page 19)

We appreciate reviewer's great suggestion on this issue. Thank you very much.

Q3) What is the destiny or fate of the Golgi membranes that are LC3-lipidated? Are they degraded by autophagy? It did not look from the images that the Golgi became fragmented.

Responses: We agree with reviewer's comment that the destiny of Golgi membrane after LC3 lipidation should be clarified as LC3 lipidation facilitates lysosomal degradation of various membranes. To address the concern, we performed additional experiments to monitor lysosomal degradation of Golgi membranes. We have constructed tandem fluorescent-tagged TGOLN2 (TGOLN2-RFP-GFP), similar to the method of analyzing autophagic flux using tandem fluorescent-tagged LC3 (Kimura *et al.*, 2007). In this construct, GFP fluorescence disappears in acidic compartments whereas RFP fluorescence is stable (Kimura *et al.*, 2007). Since RFP-GFP is conjugated to cytosolic domain of TGOLN2, if Golgi-derived vesicles are enclosed and degraded by autophagosomes/lysosomes, RFP-only dots would be observed.

We first observed the fate of TGOLN2-RFP-GFP under niclosamide treatment and the reagent was then removed for further incubation. Results revealed that the number of RFP-only TGOLN2 dots rather decreased by niclosamide treatment, which seemed to result from post-Golgi trafficking inhibition. In niclosamide wash-out conditions, the numbers of RFP-only dots were not changed compared to vehicle treatment (**New Fig. EV2A**). When we also examined protein levels of *trans*-Golgi network, TGOLN2 and Golgin-97, with immunoblotting, we found no significant difference in TGOLN2 and Golgin-97 levels between the treatment and wash-out conditions of niclosamide/AMDE-1 (**New Fig. EV2B**). While the amount of TGOLN2 protein increased with chloroquine treatment in wash-out condition for 16 h, considering **New Fig. EV2C**, this increase is likely due to the inhibition of TGOLN2 turnover in basal conditions rather than effects of niclosamide and AMDE-1. Together, we believe that LC3-lipidated Golgi membranes are not degraded through lysosomes. We added these results. (196 line on page 9 - 211 line on page 10)

New Fig. EV2

(New Fig. EV2) LC3-lipidated Golgi membranes are not degraded by lysosomes.

(A) HeLa cells expressing TGOLN2-RFP-GFP were incubated with 10 μ M niclosamide for 6 h and niclosamide was then removed for additional 16 h. Nuclei were stained by Hoechst dye 33342. Scale bar, 10 μ m (left). Colocalization of GFP and RFP in A was quantified with Fiji and represented as Pearson's correlation coefficient. Bars represent mean \pm s.d. ($n = 3, 33 \sim 47$ cells per experiment, one-way ANOVA followed by Tukey's multiple comparisons test) (right). (B,C) HeLa cells were incubated with 10 μ M niclosamide or 10 μ M AMDE-1 for 6 h. Niclosamide and AMDE-1 were removed for additional 6 h or 16 h in the presence or absence 50 μ M chloroquine. Cells were subjected to immunoblot analysis.

In Golgi-LC3 lipidation, we found that V-ATPase inhibitors reversed LC3 accumulation (Fig. 4A in the manuscript). In general, CASM inhibition by V-ATPase inhibitors is thought to operate at the stage of V0-V1 subunit assembly before LC3 lipidation occurs, rather than inducing the de-lipidation process itself. In addition, V-ATPase inhibition blocks lysosomal activity and degradation. Therefore, we believe that the reversal of LC3 accumulation following V-ATPase inhibition results from de-lipidation of the previously lipidated LC3 after the inhibition of pre-lipidation stage and de-lipidation itself is independent of V-ATPase inhibitors. Collectively, we hypothesize that after Golgi-LC3 lipidation, LC3 would be delipidated and dissociated from the Golgi membranes rather than destruction of LC3-lipidated Golgi membranes through lysosomes.

We appreciate reviewer for constructive and valuable comments that improve this manuscript.

Our response to the comments of Reviewer #3

Although recent studies have shown that ATG8/LC3 is lipidated with PS in the Golgi (CASM) upon Golgi stress, its molecular mechanism has been unknown. In this manuscript, the authors found that overexpression of DLK1 induces Golgi-LC3 lipidation via ATG12-ATG5-ATG16L1 complex, resulting in nuclear translocation of TFE3 (a transcription factor regulating the Golgi stress response). In addition, they revealed that inhibition of post-Golgi trafficking precedes Golgi-LC3 lipidation. Interestingly, Golgi-LC3 lipidation is cytoprotective against Golgi stress via transcriptional activation of Golgi-related genes by TFE3. From these observations, the authors concluded that non-autophagic Golgi-LC3 lipidation activates the TFE3 pathway of the Golgi stress response to cope with Golgi dysfunction.

The reviewer thinks that their data are ample and clear to support their conclusion, and that the manuscript would become suitable for publication in the EMBO Journal after minor revision.

<Critiques>

Q1) How does overexpression of DLK1 cause Golgi stress and Golgi-LC3 lipidation? What is the known function of DLK1?

Responses: First, we apologize for lacking detailed explanation about DLK1. DLK1 is a non-canonical Notch ligand composed of six tandem EGF-like repeats and single transmembrane domain, which inhibits Notch signaling via interaction with NOTCH1 at the plasma membrane (Baladrón *et al*, 2005). DLK1 is expressed in most tissues during embryonic development, but its expression becomes restricted to endocrine tissues and immature progenitor cells (Pittaway *et al*, 2021). DLK1 is involved in adipogenesis and multiple tissue development (Falix *et al*, 2012). Although exact mechanisms are not fully understood, DLK1 is upregulated in a wide range of cancers and seems to have active roles in tumorigenesis (Pittaway *et al.*, 2021). Most of these functions occur when DLK1 is located on the plasma membrane. Therefore, we believe that Golgi-LC3 lipidation under DLK1 overexpression is unrelated to the aforementioned functions of DLK1. The phenomenon we observed is likely due to Golgi stress caused by DLK1 overexpression, which prevents DLK1 from arriving at the plasma membrane. We believe that Golgi apparatus localization of large amounts of DLK1 provides primarily triggering point for generating Golgi stress. We added detailed explanation about DLK1 in introduction/discussion. (110 - 112 lines on page 6, 369 - 376 lines on page 17)

Q2) Does overexpression of DLK1 inhibit post-Golgi protein trafficking? How about overexpression of other secretory proteins? The reviewer wonder if overexpression of secretory proteins such as DLK1 overwhelms Golgi function, which causes Golgi stress (inhibition of post-Golgi protein trafficking) and finally TFE3 activation.

Responses: We agree with reviewer's concern that we should examine Golgi-LC3 lipidation under overexpression of other secretory proteins. As DLK1 is one of the Notch ligands, we tested overexpression effects of other Notch ligands DLL1 and DLL3, which have tandem EGF-like repeats and single transmembrane domain similar to DLK1 (Falix *et al.*, 2012). Unlike DLK1, overexpression of DLL1 or DLL3 neither accumulated GFP-LC3B (**New Fig. EV1D**) nor induced GFP-LC3B-II conversion (**New Fig. EV1E**). In addition we have always concerned this point, too and thus examined overexpression effects of other membrane and secretory proteins on GFP-LC3B accumulation on Golgi apparatus, as stated in the text. Until now, there was no such activity as DLK1.

With retention using selective hooks (RUSH) system, in which biotin addition releases SBP-GFP-GPI from the ER to the plasma membrane through the Golgi apparatus, overexpression

of DLK1 only, but not DLL1 and DLL3, blocked biotin-induced release of SBP-GFP-GPI (Reviewer Fig. 4). Together, these results demonstrate that Golgi-LC3 lipidation occurs specifically in response to overexpression of DLK1 rather than most secretory/membrane proteins. We added these results. (133 - 138 lines on page 7)

New Fig EV.1

(New Fig. EV1C,D) DLL1 and DLL3 do not induce Golgi-LC3 lipidation.

(D) HeLa cells expressing pcDNA3-HA (Ctrl), DLK1-HA, DLL1-Myc and DLL3-Myc together with GFP-LC3B were observed by fluorescence microscopy (left). Scale bar, 20 μm. The percentages of cells with GFP-LC3B clusters are represented as mean ± s.d. (n = 3, one-way ANOVA followed by Tukey's multiple comparisons test) (right).

(E) Immunoblot analysis of HeLa cells expressing pcDNA3-HA (Ctrl), DLK1-HA, DLL1-Myc and DLL3-Myc together with GFP-LC3B (left). Relative signals of GFP-LC3B II and GFP-LC3B I on the blots are represented as mean ± s.d. (n = 3, one-way ANOVA followed by Tukey's multiple comparisons test) (right).

Reviewer Fig. 4

(Reviewer Fig. 4) DLL1 and DLL3 do not inhibit post-Golgi trafficking.

HeLa cells expressing pcDNA3-HA (Ctrl), DLK1-HA, DLL1-Myc and DLL3-Myc together with STR-KDEL-SBP-GFP-GPI were treated with 100 μM biotin and observed by fluorescence microscopy. Scale bar, 20 μm.

Q3) It would be better to describe authors speculation on molecular mechanism how Golgi-lipidation of LC3 induces dephosphorylation of TFE3.

Responses: We thank reviewer's comment which is an important remaining question. We first described our hypothesis in discussion section at the initial submission as follows; "As TFE3 does not directly bind to LC3 (Fig. EV7), Golgi-LC3 lipidation may activate Golgi stress-specific phosphatases to dephosphorylate TFE3 for its nuclear translocation. Considering TFE3 dephosphorylation by calcineurin under ER stress (Martina et al., 2016) and calcium efflux by LC3 lipidation under lysosomal stress (Nakamura et al., 2020), LC3 lipidation may trigger calcium efflux from the Golgi apparatus, leading to TFE3 dephosphorylation via calcineurin." During this revision, we performed several experiments to support our previous hypothesis as also seen in the 2nd question of Reviewer #2. We have measured calcium flux under Golgi stress but found no significant differences under the stress of post-Golgi trafficking inhibition (New Appendix Fig. S5A-C). Further research

on Golgi stress sensor and kinase and phosphatase of TFE3 pathway will provide insights how LC3 lipidation regulates TFE3. Besides calcium, the release of other Golgi-resident metal ions and its effect on the unidentified TFE3 phosphatase, and role of LC3 lipidation in metal ions release could be potentially addressed. We have revised discussion section according to this suggestion. (420 - 425 lines on page 19)

We appreciate all these beautiful and constructive suggestions. Thank you very much.

References

- Baladrón V, Ruiz-Hidalgo MJ, Nueda ML, Díaz-Guerra MJ, García-Ramírez JJ, Bonvini E, Gubina E, Laborda J (2005) dlk acts as a negative regulator of Notch1 activation through interactions with specific EGF-like repeats. *Exp Cell Res* 303: 343-359
- Banerjee S, Kane PM (2017) Direct interaction of the Golgi V-ATPase a-subunit isoform with PI(4)P drives localization of Golgi V-ATPases in yeast. *Mol Biol Cell* 28: 2518-2530
- De Matteis MA, Wilson C, D'Angelo G (2013) Phosphatidylinositol-4-phosphate: the Golgi and beyond. *Bioessays* 35: 612-622
- Egorov MV, Capestrano M, Vorontsova OA, Di Pentima A, Egorova AV, Mariggio S, Ayala MI, Tetè S, Gorski JL, Luini A *et al* (2009) Faciogenital dysplasia protein (FGD1) regulates export of cargo proteins from the golgi complex via Cdc42 activation. *Mol Biol Cell* 20: 2413-2427
- Falix FA, Aronson DC, Lamers WH, Gaemers IC (2012) Possible roles of DLK1 in the Notch pathway during development and disease. *Biochim Biophys Acta* 1822: 988-995
- Gammoh N (2020) The multifaceted functions of ATG16L1 in autophagy and related processes. *J Cell Sci* 133
- Kellokumpu S (2019) Golgi pH, Ion and Redox Homeostasis: How Much Do They Really Matter? *Front Cell Dev Biol* 7: 93
- Kimura S, Noda T, Yoshimori T (2007) Dissection of the autophagosome maturation process by a novel reporter protein, tandem fluorescent-tagged LC3. *Autophagy* 3: 452-460
- Mahon MJ (2011) pHluorin2: an enhanced, ratiometric, pH-sensitive green fluorescent protein. *Adv Biosci Biotechnol* 2: 132-137
- Mitra C, Kane PM (2023) Human V-ATPase a-subunit isoforms bind specifically to distinct phosphoinositide phospholipids. *bioRxiv*
- Pittaway JFH, Lipsos C, Mariniello K, Guasti L (2021) The role of delta-like non-canonical Notch ligand 1 (DLK1) in cancer. *Endocr Relat Cancer* 28: R271-r287
- Serra-Peinado C, Sicart A, Llopis J, Egea G (2016) Actin Filaments Are Involved in the Coupling of V0-V1 Domains of Vacuolar H⁺-ATPase at the Golgi Complex. *J Biol Chem* 291: 7286-7299

Taniguchi M, Nadanaka S, Tanakura S, Sawaguchi S, Midori S, Kawai Y, Yamaguchi S, Shimada Y, Nakamura Y, Matsumura Y *et al* (2015) TFE3 is a bHLH-ZIP-type transcription factor that regulates the mammalian Golgi stress response. *Cell Struct Funct* 40: 13-30

Uggeri J, Gatti R, Belletti S, Scandroglio R, Corradini R, Rotoli BM, Orlandini G (2004) Calcein-AM is a detector of intracellular oxidative activity. *Histochem Cell Biol* 122: 499-505

Dear Yong-Keun,

Thank you for addressing the reviewers' comments in a revised version of this manuscript. The manuscript was sent back to all three reviewers; their comments are pasted below. In my opinion, you have addressed all concerns satisfactorily; I would though like you to address comments Q2 and Q4 Reviewer #1 in the discussion. Before I can finally accept the manuscript, there are also some remaining editorial points which need to be addressed. In this regard, would you please:

- acknowledge grants RS-2023-KH134817 and NRF-2022R1A2B5B03001249 in our online submission system, and grant from the Korea Health Industry Development Institute (KHIDI) in the manuscript,
- remove the author credit section from the manuscript,
- upload the reagent table as a 'Reagent Table' file in .doc format (there is a template in our guide to authors if this is helpful),
- save Source data files as one figure/folder as .zip files. (E.g. all the Source data files for figure 1 need to be saved in a single folder and this needs to be zipped and then uploaded as "SD figure 1.zip" file),
- provide legends for figures 5b-c in a sequential manner (legend for figure 5c is currently provided before legend of figure 5b),
- provide exact p values in the legends of figures 1g; 2c-d; 3c-e; 4b, g; 5c-d; EV 1d; EV 2a; EV 3a-b; EV 5b,
- define the nature of entity for 'n' in the legends of figures 2b; 5g-h; 6a-c; EV 1b, e; EV 3c-d; EV 4b-c; EV 5b-c, and
- correct the section order as follows: title page with complete author information, abstract, keywords, introduction, results, discussion, materials & methods, data availability section, acknowledgements, disclosure and competing interests statement, references, main figure legends, tables, expanded figure legends.

I look forward to receiving these changes. EMBO Press is an editorially independent publishing platform for the development of EMBO scientific publications.

Best wishes,

William

William Teale, PhD
Editor
The EMBO Journal
w.teale@embojournal.org

We realize that it is difficult to revise to a specific deadline. In the interest of protecting the conceptual advance provided by the work, we recommend a revision within 3 months (11th Nov 2024). Please discuss the revision progress ahead of this time with the editor if you require more time to complete the revisions. Use the link below to submit your revision:

Referee #1:

While the authors substantially improved their manuscript, there are still some critical aspects that need to be addressed:

Q1: If no suitable antibody for image analysis is available, the authors should consider biochemical approaches to enrich the Golgi (e.g. by differential centrifugation or organelle IP) and probe for the presence of ATG16L1 and ATP6V1A at the Golgi.

Q2: Given that ATG16L1 lacking the CCD is expressed in substantial lower amounts, the authors cannot claim that "coiled-coil domain deletion also completely abolished ATG16L1 localization to the Golgi apparatus, suggesting that dimerization via coiled-coil domain could be essential for this localization." (line 242). The authors should level the expression levels of WT and Delta CCD ATG16L1 or delete this conclusion.

Q3: Given the problematic experimental setup in Figure 5F-H, the authors should consider using stable cell lines inducibly expressing DLK1. This system would avoid artefacts due to transient transfect and would allow to level DLK1 expression between parental and ATG16L1 KO cells (via clonal selection).

Q4: While the authors indeed "added detailed explanations in result section of the manuscript. (335 - 340 lines on page 15)", a discussion of this unexpected result is missing.

Referee #2:

In this revised version of their paper the authors have done a very good job addressing my comments. The additional data added during revision has clarified several issues and improved the paper. In my view this paper is acceptable for publication in the journal.

Referee #3:

The authors have thoroughly revised the manuscript, which is now suitable for publication in the journal of EMBO Journal.

We appreciate editor and reviewers for their kind and thoughtful consideration of our revised manuscript. Reviewers 2 and 3 were satisfied with our revision and reviewer 1 had some comments. As requested by the editor, we have addressed comments Q2 and Q4 of reviewer #1 in result (248-250 lines on page 11) and discussion (427-431 lines on page 20). We have also addressed all of the remaining editorial points.

Dear Prof. Jung,

I am pleased to inform you that your manuscript has been accepted for publication in the EMBO Journal.

Congratulations! I am really pleased to see this work in The EMBO Journal.

Best wishes,

William Teale

William Teale, PhD
Editor
The EMBO Journal
w.teale@embojournal.org
